# The ciliary kinesin KIF7 controls the development of the cerebral cortex by acting differentially on SHH signaling in dorsal and ventral forebrain

**María Pedraza[1†§], Valentina Grampa[1†], Sophie Scotto-Lomassese[1†], Julien Puech[1], Aude Muzerelle[1], Azka Mohammad[1], Sophie Lebon[2], Nicolas Renier[3], Christine Metin[1‡], Justine Masson[1,2*‡]**

[1]INSERM, UMR-S1270, Institut du Fer à Moulin, Sorbonne Université, Paris, France; [2]Université Paris Cité, NeuroDiderot, Inserm UMR1141, Paris, France; [3]Institut du Cerveau et de la Moelle Épinière, Laboratoire de Plasticité Structurale, Sorbonne Université, INSERM U1127, CNRS UMR7225, Paris, France

**\*For correspondence:**
justine.masson@inserm.fr

†These authors contributed equally to this work
‡These authors also contributed equally to this work

**Present address:** §Sharestist, Valencia, Spain

**Competing interest:** The authors declare that no competing interests exist.

## eLife Assessment

This **important** study provides **convincing** evidence that the Kinesin protein family member KIF7 regulates the development of the cerebral cortex and its connectivity and the specificity of Sonic Hedgehog signaling by controlling the details of Gli repressor vs activator functions. This study provides new insights into general aspects of cortical development.

**Abstract** Mutations of *KIF7*, a key ciliary component of Sonic Hedgehog (SHH) pathway, are associated in humans with cerebral cortex malformations and clinical features suggestive of cortical dysfunction. KIF7 regulates the processing of GLI-A and GLI3-R transcription factors in a SHH-dependent manner both in humans and in mice. Here, we examine the embryonic cortex development of a mouse model that lacks the expression of KIF7 (*Kif7*−/−). The cortex is composed of principal neurons generated locally in the dorsal telencephalon where SHH expression is low and inhibitory interneurons (cIN) generated in the ventral telencephalon where SHH expression is high. We observe a strong impact of *Kif7* deletion on the dorsal cortex development whose abnormalities resemble those of GLI3-R mutants: subplate cells are absent, the intermediate progenitor layer and cortical plate do not segregate properly, and corticofugal axons do not develop timely, leading to a delayed colonization of the telencephalon by thalamocortical axons. These structural defects alter the cortical distribution of cIN, which moreover exhibit intrinsic migration defects and cortical trajectories resembling those of cyclopamine-treated cIN. Our results show that *Kif7* deletion impairs the cortex development in multiple ways, exhibiting opposite effects on SHH pathway activity in the developing principal neurons and inhibitory interneurons.

## Introduction

The primary cilium is a tiny microtubule-based organelle present on the surface of nearly all mammalian cell types, including neurons, which functions as a signaling hub and transduces several signaling pathways comprising Sonic Hedgehog (SHH), Wnt, Delta/Notch, and mTOR pathways (*Park et al., 2019*). Primary cilium dysfunction causes pleiotropic diseases named ciliopathies. KIF7 is a ciliary kinesin responsible for the trafficking and the positive and negative regulation of the GLI transcription

factors in the primary cilium of mammals (*Haycraft et al., 2005*). Loss of function of KIF7 in the mouse has shown that KIF7 regulates SHH signaling by acting downstream of Smoothened (Smo) and upstream of GLI2 and GLI3 (*Cheung et al., 2009*; *Endoh-Yamagami et al., 2009*; *Liem et al., 2009*; *Ingham and McMahon, 2009*; *Pedersen and Akhmanova, 2014*). Studies in mice established that KIF7 activity is dependent on the expression level of SHH. In the absence of SHH, KIF7 localizes to the base of the primary cilium (*Liem et al., 2009*). GLI factors are phosphorylated and addressed to the proteasome at the base of the primary cilium for degradation, leading to the formation of a cleaved and stable transcriptional repressor (GLI3-R). In the presence of SHH, KIF7 accumulates at the distal tip of the primary cilium (*Endoh-Yamagami et al., 2009*; *Liem et al., 2009*) and associates with full-length GLI2/3 that become transcriptional activators (GLI-A) (*Han et al., 2019*).

Since 2011, 10 studies have identified patients carrying mutations in the *KIF7* gene responsible for ciliopathies classified according to clinical features as hydrolethalus, acrocallosal, Joubert, and Greig cephalopolysyndactyly syndromes (*Putoux et al., 2011*; *Putoux et al., 2012*; *Ali et al., 2012*; *Walsh et al., 2013*; *Barakeh et al., 2015*; *Ibisler et al., 2015*; *Tunovic et al., 2015*; *Asadollahi et al., 2018*; *Niceta et al., 2020*). MRI investigations revealed macrocephaly, ventricles enlargement, and corpus callosum alterations. They also showed the characteristic hindbrain abnormalities observed in most ciliopathies such as molar tooth sign (MTS) and cerebellar atrophy. Patients also presented with mild but frequent cortical malformations, as well as neurodevelopmental delay, intellectual disability, and seizures (*Putoux et al., 2012*; *Ali et al., 2012*; *Ibisler et al., 2015*; *Dafinger et al., 2011*), which indicate cortical abnormalities.

In physiological conditions, the proper activity of the cortex relies on the excitatory/inhibitory balance, i.e., on the ratio, positioning, and connectivity of excitatory glutamatergic neurons (principal neurons [PNs]) and inhibitory GABAergic interneurons [cIN], generated from progenitors in the dorsal and ventral telencephalon, respectively. The cortical neurons of dorsal and ventral origin each migrate toward the developing cortex, the so-called cortical plate (CP), according to a specific timeline allowing the establishment of proper connections in the CP.

SHH plays a central role in the forebrain patterning and differentiation. At an early embryonic stage, the ventral expression of SHH orchestrates the ventro-dorsal (*Chiang et al., 1996*; *Shimamura and Rubenstein, 1997*) and medio-lateral (*Kohtz et al., 1998*) regionalization of the mouse forebrain and the differentiation of ventral cell types (*Chiang et al., 1996*). *Shh* ablation performed after forebrain patterning alters the specification of distinct subgroups of GABAergic interneurons (*Machold et al., 2003*; *Xu et al., 2005*; *Xu et al., 2010*; *Sousa and Fishell, 2010*). Interestingly, conditional ablation of *Shh* and *Smo* in the embryonic cortex reduces the proliferation of dorsal progenitors (*Komada et al., 2008*), demonstrating a minimal SHH expression in the dorsal telencephalon, even before birth (*Dahmane et al., 2001*). Beside its role on proliferation and specification, SHH also controls the migration of interneurons to the CP (*Baudoin et al., 2012*; *Higginbotham et al., 2012*; *Komada, 2012*; *Komada et al., 2013*; *Hasenpusch-Theil and Theil, 2021*). In the embryonic telencephalon, GLI transcription factors mediate SHH signals in complex and specific ways. The three GLI factors identified in mammals are expressed in the mouse forebrain: GLI1 along the source of SHH in the ganglionic sulcus, and GLI2 and GLI3 dorsally to the SHH source (*Sousa and Fishell, 2010*; *Hui et al., 1994*; *Yu et al., 2009*). GLI1 acts uniquely as a pathway activator, whereas GLI2 and GLI3 can be processed into transcriptional activators or inhibitors in the primary cilium. However, both GLI1 and GLI2 function primarily as transcriptional activators in response to SHH activity in the ventral forebrain (*Yu et al., 2009*). Nevertheless, GLI1 and GLI2 mutants show mild phenotypes (*Park et al., 2000*; *Bai and Joyner, 2001*), and GLI2 seems required to transduce high-level SHH signals in mice (*Ding et al., 1998*). In contrast, the development of the dorsal cortex, where principal excitatory neurons differentiate, depends mainly on the expression of the GLI3-R repressor, in agreement with the low cortical expression of SHH (*Tole et al., 2000*; *Rallu et al., 2002*). Following patterning, Gli3R/A ratio remains critical for specifying the fate of cortical progenitors and regulating cell cycle kinetics (*Wang et al., 2011*; *Wilson et al., 2012*; *Hasenpusch-Theil et al., 2018*).

Previous studies investigated the mechanisms underlying the corpus callosum agenesis in patients with *KIF7* mutation using *Kif7* knockout (*Kif7−/−*) mice (*Putoux et al., 2019*) and the consequence of KIF7 knockdown on cortical neurogenesis in principal cells electroporated with *Kif7* shRNA (*Guo et al., 2015*). It remains that the influence of developmental abnormalities associated with ciliopathies on the cortical cytoarchitecture is poorly understood. Given the crucial role of KIF7 in regulating the

GLI pathways and the important role of Gli3-R and GLI2/3-A in regulating the early developmental stages of the dorsal and ventral telencephalon, respectively, we examined here the cortical development from embryonic stage 12.5 (E12.5), the establishment of long-distance projections with the thalamus and the migration of GABAergic interneurons (cIN) in *Kif7*-/- mice. The migratory behavior of *Kif7*-/- cINs was investigated by time-lapse videomicroscopy in co-cultures and in organotypic cortical slices. We also compared the migration of cINs in *Kif7*-/- cortical slices and in control slices treated with pharmacological activator and inhibitor of the SHH pathway. We moreover determined the local distribution of the SHH protein in the embryonic cortex.

Our results show developmental defects leading to permanently displaced neurons, abnormal formation of cortical layers, and defective cortical circuits that could be responsible for epilepsy and/or intellectual disability in patients carrying *KIF7* mutation (*Bakalinova, 1998*; *Putoux et al., 2012*; *Ali et al., 2012*; *Walsh et al., 2013*; *Barakeh et al., 2015*; *Ibisler et al., 2015*; *Tunovic et al., 2015*; *Asadollahi et al., 2018*; *Niceta et al., 2020*; *Putoux et al., 2019*).

## Results

### Kif7 knockout mice as a model to investigate the structural cortical defects that could lead to clinical features in patients carrying KIF7 mutation

The *KIF7* gene is located on chromosome 15 in humans and encodes a 1343 aa protein containing a kinesin motor domain and a GLI-binding domain in the N-terminal part followed by a coiled-coil region and a Cargo domain able to bind a diverse set of cargos in the C-terminal part (*Maurya et al., 2013*). *Table 1* summarizes the clinical features associated with mutations targeting either the kinesin or GLI-binding domain, the coiled-coil domain, or the N-terminal Cargo domain. Interestingly, various mutations are associated with the same clinical picture, suggesting that *KIF7* mutations, whatever their nature, could lead to protein loss of function, e.g., by altering the protein structure and solubility as proposed by *Klejnot and Kozielski, 2012*. All patients carrying a mutation in the *KIF7* gene have developmental delay (DD) and intellectual disability (ID) associated with classical defects of ciliopathies (ventricle enlargement, macrocephaly, corpus callosum agenesis, and MTS). Some patients have additional anatomical cerebral cortex defects such as poor frontal development, atrophy, pachygyria, and heterotopia (*Bakalinova, 1998*; *Putoux et al., 2011*; *Walsh et al., 2013*; *Tunovic et al., 2015*), which could participate not only in DD and ID but moreover in seizure as observed in a quarter of patients. In this context, we used a murine model in which the *Kif7* gene had been deleted (*Kif7*-/-) (*Cheung et al., 2009*) to investigate the consequence of KIF7 loss of function on the cortex development. *Kif7*-/- mice have been previously characterized as dying at birth with severe malformations, skeletal abnormalities (digits and ribs), neural tube patterning defects, including exencephaly in a third of the mutants, microphtalmia, lack of olfactory bulbs, and CC agenesis (*Cheung et al., 2009*; *Endoh-Yamagami et al., 2009*; *Putoux et al., 2019*). In the present study, the KIF7 loss of function was analyzed in a mouse strain with a mixed genetic background in which exencephaly was observed at the same frequency in *Kif7*-/- and control embryos. As previously reported, *Kif7*-/- embryos displayed microphthalmia (illustrated at E14.5, *Figure 1A*, black arrow) and polydactyly (not illustrated). Interestingly, skin laxity previously reported in ciliopathic patients (*Chen et al., 2015*; *Walczak-Sztulpa et al., 2010*) was observed in all mutants (*Figure 1A*, white arrow). Comparison of brains from *Kif7*-/- embryos and control littermates revealed the absence of olfactory bulbs in *Kif7*-/- brains (illustrated at E14.5, *Figure 1B*, white arrows on ventral face view).

### Kif7 invalidation alters the development of the cortex

The thinning of the dorsal cortex at E14.5 allowing the lateral ventricles to be seen through (*Figure 1B*, black arrow on dorsal face view) proned us to investigate the structural organization of the forebrain over time (*Figure 1C1-3*, E12.5-E16.5). Analyses performed on rostro-caudal series of frontal sections identified strong abnormalities in the cortex development. E12.5 *Kif7*-/- embryos exhibited cortical wall folding and thinning in rostro-median sections (*Figure 1C1*, arrowheads) and prominent diencephalon (*Figure 1C1*, stars). At E14.5, cortex folding was no longer observed in *Kif7*-/- embryos which exhibited enlarged lateral ventricles (compare left and right panels in *Figure 1C2*). The cortex remained thinned in the dorsal region, and the latero-dorsal extension of the telencephalic vesicle was strongly reduced

**Table 1.** Clinical diagnosis of patients carrying a mutation in the *KIF7* gene on both alleles from the literature.
The ablated or mutated domains were identified. Clinical features associated with cortical dysfunction are listed. Among cerebral defects, those observed in the cortex are enlightened. DD, developmental delay; ID, intellectual disability; CC, corpus callosum; MTS, molar tooth sign.

| Mutation | Domain lacking, truncated, or mutation | Brain | | Ref. |
|---|---|---|---|---|
| | | Clinical feature | Anatomical defects (cortical in bold) | |
| Hom c.67C>T, p.Arg33* | No kinesin motor and no Gli binding | DD/ID | Wide ventricles - Macrocephaly CC agenesis/hypoplasia - MTS | *Putoux et al., 2011* |
| Hom c.233_234del, p.Leu78Profs*2 | Truncated kinesin motor and no Gli binding | DD/ID | Macrocephaly - CC agenesis - MTS | *Putoux et al., 2011* |
| Hom c.587dupT, p.Glu197Glyfs*19 | Truncated kinesin motor, no Gli binding | DD/ID | Macrocephaly - CC agenesis | *Putoux et al., 2011* |
| Hom c.687delG, p.Arg230Alafs*92 | Truncated kinesin motor, no Gli binding | DD/ID | Wide ventricles - Macrocephaly - CC agenesis - **Poor frontal cortical development** | *Putoux et al., 2011* |
| Hom c.217 delG, p.Ala73Profs*109 | No Gli binding | DD/ID | CC agenesis/hypoplasia - MTS | *Dafinger et al., 2011* |
| Hom c.653_662del, p.218-221del | Deletion in kinesin motor | DD/ID | Wide ventricles - Macrocephaly - CC agenesis - **Cortical atrophy** | *Walsh et al., 2013* |
| Hom c.1639_1640delinsT, p.Gly547Serfs*5 | No coiled-coil | DD/ID | Macrocephaly - MTS - CC agenesis/hypoplasia - **Temporal pachygyria** | *Putoux et al., 2011* |
| Hom c.1643dupC, p.Arg549Alafs*40 | No coiled-coil | DD/Mild ID | Macrocephaly - CC dysgenesis - Cerebellar hypotrophy | *Asadollahi et al., 2018* |
| Hom c.1643dupC, p.Arg549Alafs*40 | No coiled-coil | DD/Mild ID | Macrocephaly - CC agenesis - MTS | *Asadollahi et al., 2018* |
| Hom c.2164G>T, p.Glu722* | No coiled-coil | DD/Mild ID | Macrocephaly | *Asadollahi et al., 2018* |
| Hom c.2335G>T, p.Glu779 * | No coiled-coil | DD/ID - Seizure | Macrocephaly - CC agenesis - Cerebellar hypoplasia - Abnormal formation of the brainstem | *Ibisler et al., 2015* |
| Hom c.2896_2897del | Deletion in coiled-coil | DD/ID | CC dysgenesis | *Barakeh et al., 2015* |
| Hom c.3001C>T, p.Gln1001* | Truncated coiled-coil | DD/ID | Wide ventricles - Macrocephaly - CC agenesis/hypoplasia- MTS | *Putoux et al., 2011* |
| Hom c.529+2T>C | Mutation in kinesin motor | DD | MTS | *Putoux et al., 2011* |
| c.2593–3C>G / /c.3001C>T, p.Gln1001* | Mutation in coiled-coil // Truncated coiled-coil | DD/ID - Seizure | Macrocephaly - CC agenesis | *Asadollahi et al., 2018* |
| c.1019dupT, p.Asn341Glnfs*122// c.3331C>T, p.Arg1111* | Truncated kinesin motor and no Gli binding // Truncated coiled-coil | DD | CC agenesis | *Putoux et al., 2012* |
| Hom c2593C>G / /c3001C>T, p.? // p.Gln1001* | Mutation in coiled-coil // Truncated Coiled-coil | DD/ID | Macrocephaly - CC hypoplasia | *Asadollahi et al., 2018* |
| Hom c.2272G>T | Mutation in coiled-coil | DD/ID - Seizure | CC agenesis | *Barakeh et al., 2015* |
| Hom c.3179A>G | Mutation in coiled-coil | DD/ Very mild ID | Macrocephaly – MTS - CC agenesis/hypoplasia - **Temporo-parietal atrophy** | *Bakalinova, 1998; Ali et al., 2012* |
| c.461G>A / / c.2959 G>A | Mutation in coiled-coil | DD/ID - Seizure | Hydrocephalus - Cerebellar hypoplasia - CC agenesis/hypoplasia - **Pachygyria - Heterotopia** | *Tunovic et al., 2015* |
| c.3365C>G / / c.2482G>A | Mutation in coiled-coil | DD/ID - Seizure – Ataxia | CC agenesis/hypoplasia | *Tunovic et al., 2015* |
| Hom c2593-3C>G | Mutation in coiled-coil | DD/Mild ID | Macrocephaly - CC hypoplasia Large asymmetrical cisterna magna | *Asadollahi et al., 2018* |

*Table 1 continued on next page*

*Table 1 continued*

| Mutation | Domain lacking, truncated, or mutation | Brain | | Ref. |
| | | Clinical feature | Anatomical defects (cortical in bold) | |
| --- | --- | --- | --- | --- |
| Hom c.3331C>T | Mutation in coiled-coil | DD/ID | CC agenesis - MTS | *Barakeh et al., 2015* |

caudally. These defects were maintained in E16.5 embryos (*Figure 1C3*), and the hippocampus did not develop properly (*Figure 1C3*, caudal section). A quantitative analysis performed in E14.5 brains (*Figure 1C4*) confirmed the ventriculomegaly (upper left graph) and the cortex thinning (upper right graph) in the rostral and median sections despite unchanged total width and height of the telencephalon (*Figure 1C4*, lowers graphs). KIF7 depletion in E10.5-E11.5 mouse embryos was reported to alter the dorso-ventral patterning of the medial forebrain (*Putoux et al., 2019*). The frontier of expression of GSH2 (*Figure 1D1 and D2*) and TBR2 (*Figure 1D3 and D4*), two transcription factors expressed in the ventral and dorsal telencephalon, respectively, was slightly abnormal at E13-E14 in *Kif7⁻/⁻* embryos with a small shift of GSH2(+) cells toward the ventricular angle (*Figure 1D2*) and a dispersion of some TBR2(+) cells at the pallium-subpallium boundary (PSB) (*Figure 1D4*). This suggests a near-normal dorso-ventral patterning of the telencephalic vesicles, however, with a discrete blurring of the dorsal and ventral markers at the PSB.

In *Kif7⁻/⁻* embryos, the cleavage of full-length GLI3 (GLI3-FL) into the transcriptional repressor GLI3 (GLI3-R) is decreased (*Cheung et al., 2009*; *Endoh-Yamagami et al., 2009*). Here, we analyzed the expression of GLI3-FL and GLI3-R in the cortex and in the medial ganglionic eminence (MGE) where cortical interneurons (cIN) are born. We showed minimal expression of GLI3-R and GLI3-FL in the MGE compared to the cortex (*Figure 2A*) and a strong decrease in the cleavage of GLI3-FL in GLI3-R only in the cortex of *Kif7⁻/⁻* embryos (*Figure 2A and B*).

The structural defects described above convinced us to examine the cellular organization of the cortex of *Kif7⁻/⁻* embryos. At E12.5, the whole cortex is proliferative. From stage E14.5, the mouse cortex can be described as a stack of three specialized domains: (i) the upper/superficial cortical layers containing post-mitotic cells generated in the proliferative zones of the cortex, with TBR1(+) post-mitotic neurons located in the CP in-between the subplate (SP) and marginal zone (MZ) MAP2-expressing cells; (ii) the deep proliferative layers, ventricular and subventricular zones (VZ and SVZ, respectively), located along the lateral ventricle; and (iii) the intermediate zone (IZ) located between the CP and the proliferative layers, which hosts the radially and tangentially migrating neurons and growing axons. At the end of the embryonic development, the proliferative layers are strongly reduced due to neurogenesis decrease, whereas the cortical thickness had increased due to CP delamination. At E14.5, the TBR1(+) CP cells appeared more clustered in *Kif7⁻/⁻* embryos than in wild-type (WT) embryos where they formed a rather regular, radially organized CP (compare left and right panels in *Figure 3A1, A2*). The thickness of the upper MAP2(+) MZ layer was extremely irregular in *Kif7⁻/⁻* embryos, and MAP2(+) SP cells were missing in the dorsal cortex (*Figure 3A1 and A2*, white arrows indicate the limit of distribution of MAP2(+) SP cells). In E16.5 and E18.5 embryos, the CP was obviously thinner in the mutant as compared to the WT (*Figure 3B and C*), even though CTIP(+) cells differentiated normally in V-VI layers (*Figure 3C*). The CP thinning appeared even more pronounced in the caudal than in the median and rostral cortex. We then analyzed the proliferative layers focusing on TBR2(+) intermediate progenitors (IPs) which normally form a dense layer in the cortical SVZ (*Figure 3D1*, left). The TBR2(+) layer constantly showed an abnormal positioning in the dorsal cortex of *Kif7⁻/⁻* embryos. TBR2(+) cells indeed reached the brain surface (*Figure 3D1*, arrow) where they mixed up with post-mitotic TBR1(+) cells instead of keeping a minimal distance with TBR1 cells as observed in control brains (*Figure 3D2*, arrow). As a consequence, IPs and post-mitotic neurons no longer segregated in the dorsal cortex of *Kif7⁻/⁻* embryos where no IZ was observed (*Figure 3D2*, arrow). Accordingly, to the thinning of the cortex starting at E12.5, the TBR2(+) layer was thinner at all development stages analyzed (*Figure 3E and F*). In contrast, the abnormal positioning of TBR2(+) cells at the brain surface in the dorsal cortex of E14.5 embryos was no longer observed at E16.5 (*Figure 3F*), suggesting either a transient structural disorganization and/or DD of the dorsal cortex in *Kif7⁻/⁻* embryos at E14.5. The TBR2 staining, moreover, revealed focal heterotopia in the dorsal or lateral cortex of E14.5 *Kif7⁻/⁻* embryos, either at the ventricular or pial surface (*Figure 3—figure supplement 1*).

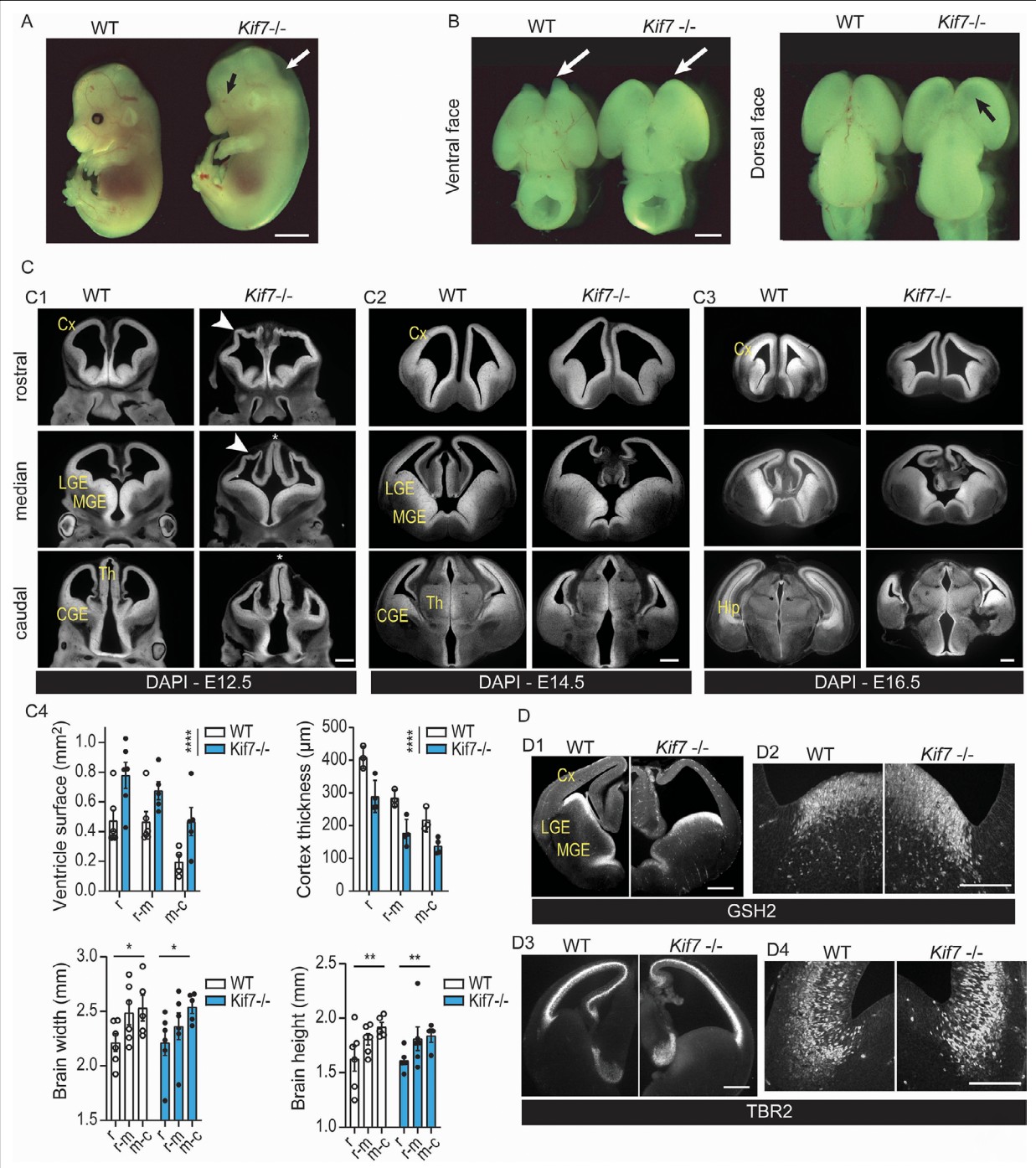

**Figure 1.** *Kif7* deletion alters cortical development on latero-medial and rostro-caudal axes. (**A**) *Kif7*-/- embryos are microphtalmic (black arrow) and exhibit skin laxity (white arrow). (**B**) External examination of the brain reveals the lack of olfactory bulbs (white arrows, left panel) and the thinning of the dorsal telencephalon (black arrow, right panel). (**C**) DAPI staining of rostro-caudal series of coronal sections at embryonic stage 12.5 (E12.5) (**C1**), E14.5, (**C2**), and E16.5 (**C3**) illustrates the anatomical defects of *Kif7*-/- embryonic brains quantified at E14.5 in C4. The ventricles of *Kif7*-/- embryos are strongly enlarged (upper left graph; WT, n=4–6, *Kif7*-/-, n=5–6 depending on the rostro-caudal level), their cortical thickness strongly decreased (upper right graph; WT, n=3; *Kif7*-/-, n=4), resulting in minimal brain width (lower left graph; WT, n=5–6; n=5–6 for *Kif7*-/- depending on the rostro-caudal level) and height (lower right graph; WT, n=5–6; *Kif7*-/-, n=4–6 depending on the rostro-caudal level) changes. Statistical significance was tested by two-way ANOVA or mixed model (GraphPad 8.1.0). For ventricle surface and cortex thickness, the mixed model reveals a genotype effect (p<0.0001). For brain width and height, no genotype effect was observed, but a significant effect of the rostro-caudal level on brain width (p=0.0441) and height (p=0.0092). (**D**) The pallium-subpallium boundary identified by the limit of expression of ventral (GSH2, **D1, D2**) and dorsal (TBR2, **D3, D4**) telencephalic markers is less precisely defined and slightly shifted to the ventricular angle in E13-E14 *Kif7*-/- embryos compared to wild-type embryos (epifluorescent low

*Figure 1 continued on next page*

Figure 1 continued

magnification, D1, D3; confocal high magnification images, **D2, D4**). Graphs in C4 represent the means and SEM. Cx, cortex; Hip, hippocampus; LGE, lateral ganglionic eminence; MGE, median ganglionic eminence; CGE, caudal ganglionic eminence; Th, thalamus. Scale bars, 2 mm (**A**), 1 mm (**B**), 500 μm (**C, D1, D3**), 150 μm (**D2, D4**).

## The loss of Kif7 alters the connectivity between the cortex and the thalamus

The IZ hosts migrating neurons and the growing cortical and thalamic projections (*Price et al., 2006*). After reaching the CP by radial migration, post-mitotic neurons extended pioneer axons oriented tangentially in the IZ and directed to the PSB (*Hatanaka and Yamauchi, 2013*; *Kon et al., 2017*). We thus examined whether the structural defects of the CP and SP and the lack of IZ in the dorsal cortex of *Kif7⁻/⁻* embryos did associate with developmental abnormalities of corticofugal projections. We labeled corticofugal axons by inserting small crystals of DiI in the CP of E14.5 paraformaldehyde (PFA) fixed brains. After DiI had diffused along corticofugal axons, we analyzed labeled axons on coronal sections. In both control and *Kif7⁻/⁻* brains, corticofugal axons extended below the CP toward the PSB (*Figure 4A*). According to the latero-medial gradient of cortical development, DiI injections in the dorsal cortex of control and *Kif7⁻/⁻* brains (*Figure 4A1*) labeled much less axons than injections in the lateral cortex (*Figure 4A2 and A3*). Remarkably, the cortical bundles labeled from similar cortical regions were always much smaller and shorter in *Kif7⁻/⁻* than in WT brains (compare right and left columns in *Figure 4A*). Dorsal injections labeled large bundles crossing the PSB in WT brains, but only a few axons reached the PSB in *Kif7⁻/⁻* brains

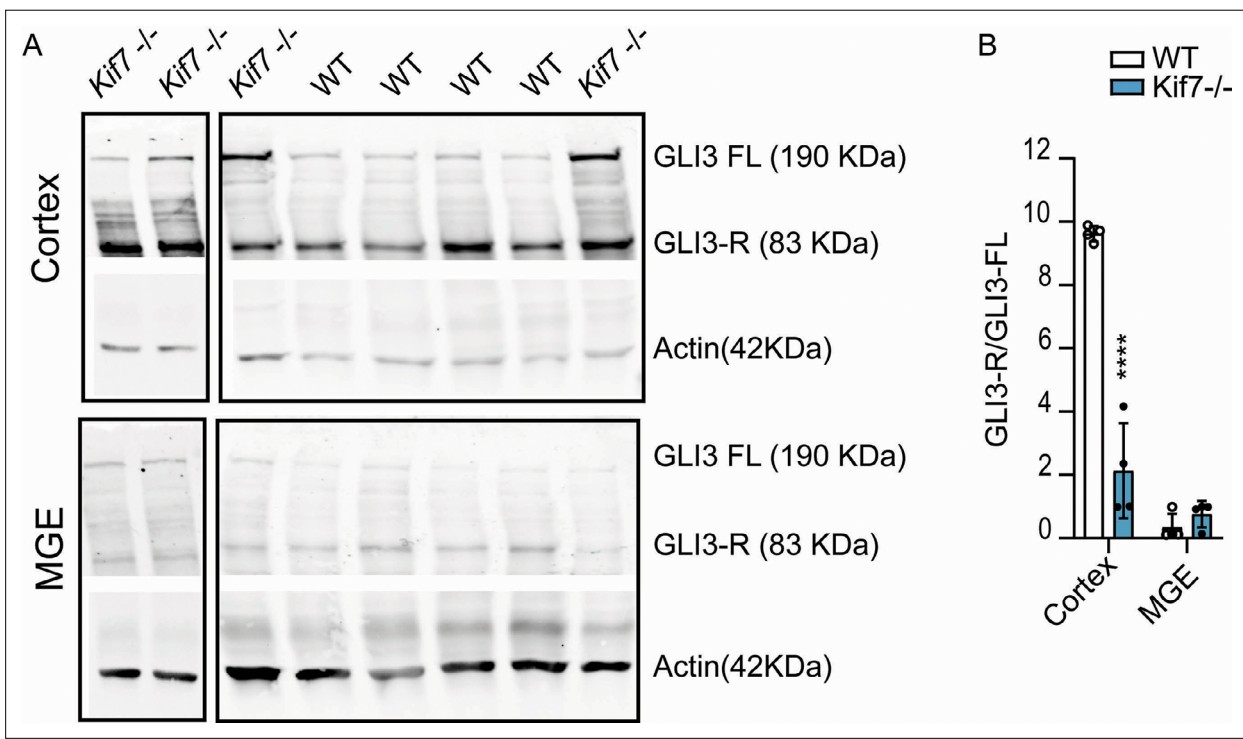

**Figure 2.** Western blot analysis on the cortex and medial ganglionic eminence (MGE) of wild-type (WT) and *Kif7⁻/⁻* embryos at embryonic stage 14.5 (E14.5). (**A**) The cleaved form of GLI3 (GLI3-R at 83 KDa) and the full-length GLI3 (GLI3-FL at 190 KDa) are more abundant in the cortex compared to the MGE (see actin band intensity for protein loading). (**B**) The ratio Gli3-R/Gli3-FL is lower in the MGE than in the cortex of WT animals and is significantly decreased only in the cortex of *Kif7⁻/⁻* brains compared to control. Two-way ANOVA reveals significant interaction between brain structure and genotype (WT, n=4; *Kif7⁻/⁻*, n=4; p=0.002), and multiple comparisons show a statistical difference between genotype only in the cortex (****, p<0.0001). Graph represents the means and SEM.

The online version of this article includes the following source data for figure 2:

**Source data 1.** Original files for western blot analysis displayed in *Figure 2*.

**Source data 2.** PDF file containing original western blots for *Figure 2*, indicating the genotype of samples and relevant bands.

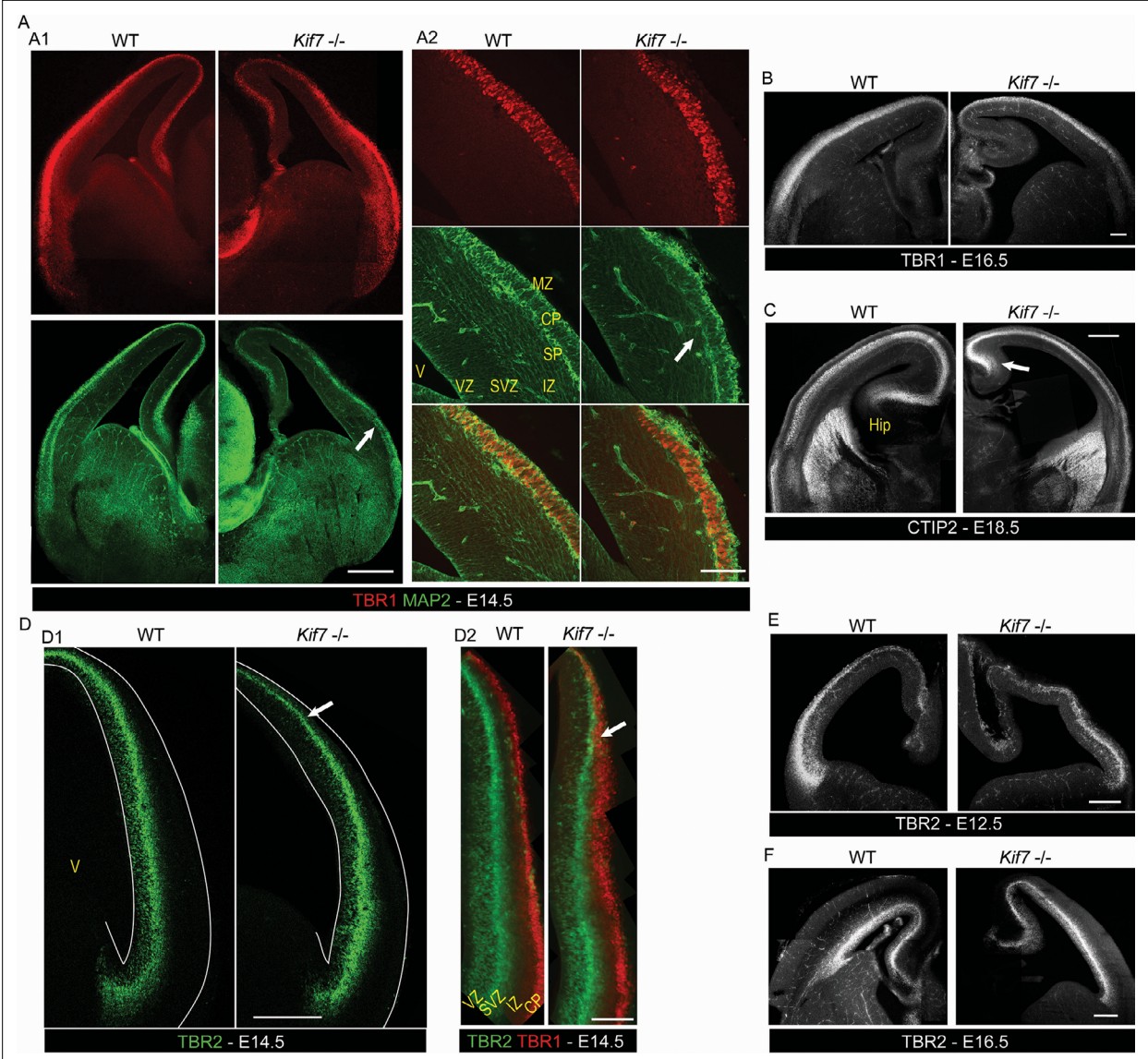

**Figure 3.** Histological alterations in the developing cortex of *Kif7⁻/⁻* embryos. (**A–C**) Immunostaining of cortical post-mitotic layers in embryonic stage 14.5 (E14.5) (**A**), E16.5 (**B**), and E18.5 (**C**) in median coronal sections representative of five wild-type (WT) and *Kif7⁻/⁻* embryos imaged on epifluorescence (**A1, B, C**) or confocal (**A2**) microscopes. At E14.5 (**A1, A2**), the TBR1(+) staining (red) of the cortical plate is more clustered in *Kif7⁻/⁻* than in WT embryos, and the MAP2(+) staining (green) of the subplate is absent in the dorsal cortex of E14.5 *Kif7⁻/⁻* embryos (white arrow, right column). The post-mitotic layers remain thinner in *Kif7⁻/⁻* embryos at later embryonic stages as illustrated by TBR1 staining at E16.5 (**B**) and CTIP1 staining at E18.5 (**C**) that specifically labels the deeper cortical layers (**V–VI**). Moreover, the hippocampus is underdeveloped in the mutant (white arrow) (**C**). (**D–F**) Immunostaining of TBR2(+) proliferative layer. At E14.5 (**D1, D2**), the TBR2(+) layer (green) of secondary progenitors appears disorganized in the lateral cortex of the *Kif7⁻/⁻* embryos (white arrowhead in D1) and reaches the brain surface in the dorsal cortex of *Kif7⁻/⁻* embryo where it intermingles with post-mitotic TBR1(+) cells (D2, red) (white arrows). In E12.5 *Kif7⁻/⁻* embryo (**E**), the TBR2(+) layer is thin and clustered; however, its localization in the thickness of the cortex is normal. At the E16.5 stage (**F**), the TBR2(+) layer is normally positioned in the deeper layer of the cortex in both WT and *Kif7⁻/⁻* embryos. Hip, hippocampus; V, ventricle; VZ, ventricular zone; SVZ, subventricular zone; IZ, intermediate zone; CP, cortical plate; MZ, marginal zone. Scale bars: 250 µm (**A1, B, F**), 100 µm (**A2**), 200 µm (**D, E**), 500 µm (**C**).

The online version of this article includes the following figure supplement(s) for figure 3:

**Figure supplement 1.** *Kif7* deletion is associated with cortical heterotopia at embryonic stage 14.5 (E14.5).

(*Figure 4A1*). The growth of corticofugal axons appeared severely hampered in the dorsal cortex of *Kif7⁻/⁻* embryos. Lateral injections in WT brains labeled anterogradely corticofugal axons that reached the embryonic striatum and continued medially in the internal capsule (IC) and labeled retrogradely thalamic projections (*Figure 4A3*, left panel). Lateral injections in *Kif7⁻/⁻* embryos

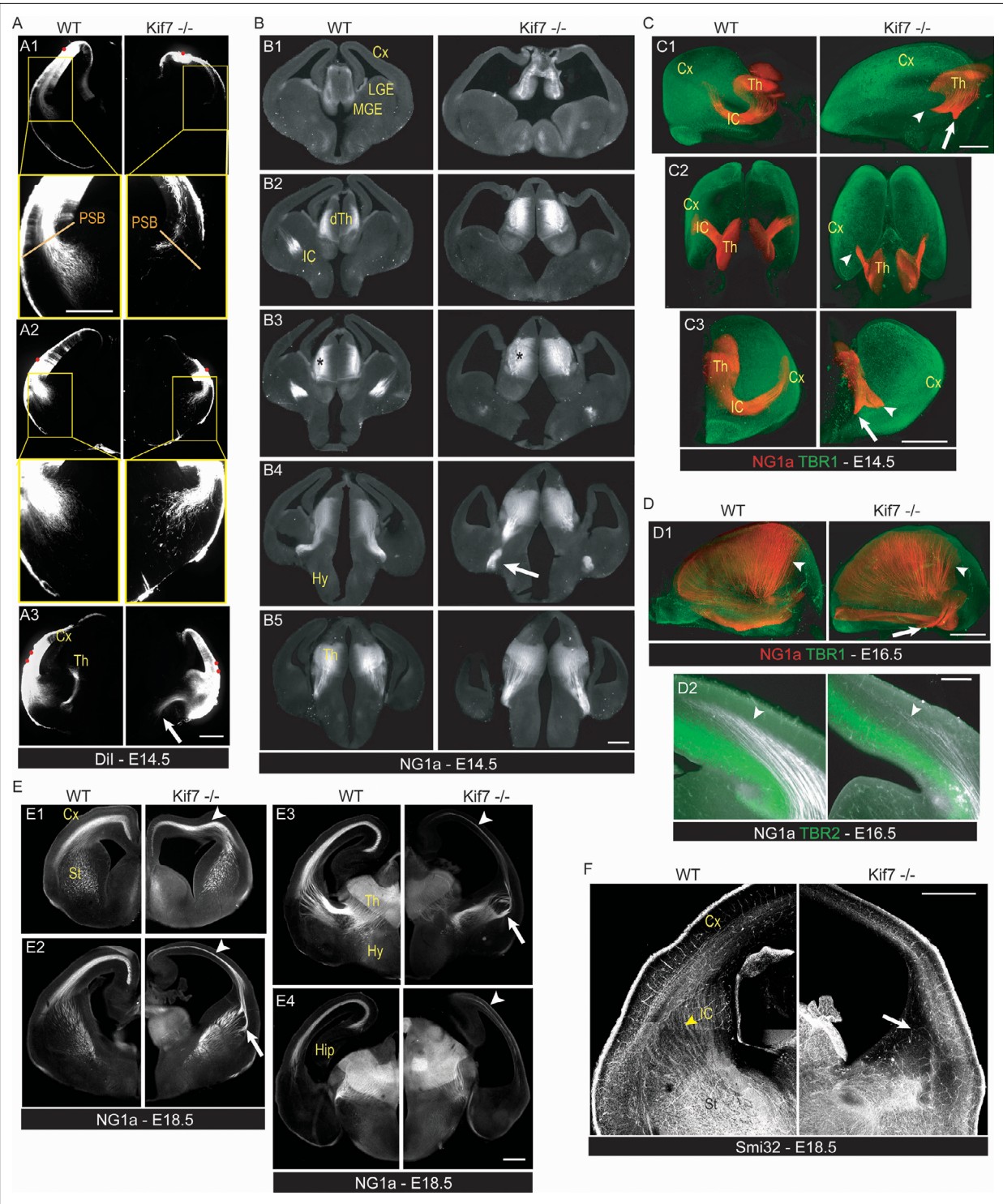

**Figure 4.** *Kif7* deletion disrupts the connectivity between the cortex and the thalamus. (**A**) Panels illustrate the corticofugal projections labeled by DiI crystal (red dots) positioned in the dorsal (**A1**) and lateral (**A2, A3**) cortex of embryonic stage 14.5 (E14.5) wild-type (WT) (left) and *Kif7*⁻/⁻ (right) embryos on vibratome sections performed 30 days after DiI placement and imaged on a macroscope. In *Kif7*⁻/⁻ embryos, fewer axons project from the dorsal (**A1**) and lateral (**A2**) cortex to the subpallium than in WT (compare enlarged views of the projections below A1 and A2). Cortical injections in *Kif7*⁻/⁻ embryos do not label thalamic axons (compare left and right panels in A3) but label a ventral projection (A3, white arrow). (**B**) Rostro-caudal series of coronal sections (**B1–B5**) immunostained with anti-Netrin G1a (NG1a) antibodies compare the trajectory of thalamocortical axons (TCA) in an E14.5 WT (left) and in a *Kif7*⁻/⁻ (right) embryo imaged on a macroscope. TCA reach the pallium-subpallium boundary of the WT embryo, whereas they are lost in the ventral forebrain of the mutant (B4, white arrow). (**C**) Representative three-dimensional reconstructions (C1, lateral; C2, horizontal; C3, coronal

*Figure 4 continued on next page*

*Figure 4 continued*

views) of WT and *Kif7⁻/⁻* E14.5 brains immunostained as a whole with NG1a (red) and TBR1 (green) antibodies that label respectively the TCA and cortical plate cells before transparization and imaging with a light sheet microscope (see *Figure 4—videos 1 and 2*). While all labeled TCA extend in the internal capsule (IC) and a significant proportion of them enter the cerebral cortex in the WT brain, TCA in the *Kif7⁻/⁻* brain split in two bundles in the basal telencephalon. A bundle stops shortly after entering the IC (white arrowheads), whereas the second bundle extends ventrally (white arrows). (**D**) At E16.5, TCA are immunostained with NG1a in WT (left) and *Kif7⁻/⁻* (right) embryos in whole brain with TBR1 antibodies before transparization and imaging (**D1**) and on coronal section with TBR2 (**D2**). The TCA extend in the cortex in *Kif7⁻/⁻* brain; however, fiber density is reduced in the median and caudal brain compared to WT (D1, white arrowhead). TCA invade the post-mitotic layers in the cortex (above TBR2 layer) to a lower extent in *Kif7⁻/⁻* brain (**D2**). In the *Kif7⁻/⁻* brain, thick bundles of NG1a(+) fibers project ventrally from the caudal telencephalon, a projection never observed in control brains (D1, white arrow). (**E**) At E18.5, rostro-caudal series of coronal sections (**E1–E4**) immunostained with NG1a antibodies compare the trajectory of TCA in WT (left) and *Kif7⁻/⁻* (right) embryos imaged on a macroscope. TCA reach the dorsal cortex in both WT and *Kif7⁻/⁻* embryos. However, the TCA projection in *Kif7⁻/⁻* (arrowheads) becomes thinner in median sections (**E2**) and almost disappear in caudal sections (**E3, E4**). Fiber trajectories in the lateral striatum are abnormal (E2, E3, arrows). (**F**) Panels illustrate the cortical projections immunolabeled by Smi32 antibodies in E18.5 WT and *Kif7⁻/⁻* brain. In the *Kif7⁻/⁻* brain, the number of fibers connecting the cortex with other brain structures is strongly reduced (arrow). Cx, cortex; dTh, dorsal thalamus; Hip, hippocampus; Hy, hypothalamus; St, striatum; IC, internal capsule; LGE, lateral ganglionic eminence; MGE, median ganglionic eminence; PSB, pallium-subpallium boundary. Scale bars: 250 µm (**A–D**), 500 µm (**E, F**).

The online version of this article includes the following video and figure supplement(s) for figure 4:

**Figure supplement 1.** Alterations of the thalamocortical projection at embryonic stage 14.5 (E14.5) in *Kif7⁻/⁻* brains.

**Figure 4—video 1.** Three-dimensional (3D) maximum intensity projection of thalamocortical axons in embryonic stage 14.5 (E14.5) wild-type (WT) brain.
https://elifesciences.org/articles/100328/figures#fig4video1

**Figure 4—video 2.** Three-dimensional (3D) maximum intensity projection of thalamocortical axons in embryonic stage 14.5 (E14.5) *Kif7⁻/⁻* brain.
https://elifesciences.org/articles/100328/figures#fig4video2

labeled anterogradely cortical axons that spread in the striatum (*Figure 4A2*, right panel) and labeled a large bundle directed to the ventral pre-optic area (*Figure 4A3*, right panel, white arrow), which was never observed in WT brains. Moreover, no thalamic axons were retrogradely labeled from the lateral cortex in *Kif7⁻/⁻* embryos.

To identify the trajectories of thalamic axons in *Kif7⁻/⁻* embryos, we immunostained coronal sections of E14.5 brains with antibodies against the Netrin G1a (NG1a), an early marker of thalamocortical axons (TCA, *Figure 4B*; *Braisted et al., 2000*). At E14.5, NG1a antibodies labeled a larger population of neurons in the dorsal thalamus of *Kif7⁻/⁻* than in WT brains (*Figure 4B*, stars). In control brains, TCA made a right angle turn to join the IC in the striatum. Then they extended to the PSB, and some axons entered the lateral cortex (*Figure 4B2 and B3*, left panel). In *Kif7⁻/⁻* embryos, most thalamic axons formed a thick bundle oriented ventrally in the basal forebrain (*Figure 4B4*, white arrow in right panel). A minor structural defect was evidenced at the telo-diencephalic junction by Pax6 staining in *Kif7⁻/⁻* brains (*Figure 4—figure supplement 1*), recalling the prethalamus defect described in the ciliary Rfx3 mutant (*Magnani et al., 2015*). Given the complex trajectory of NG1a thalamic axons, we immunostained thalamic axons in whole brains and imaged them after transparization using a light-sheet microscope. The three-dimensional reconstruction of labeled projections confirmed that thalamic axons made a sharp turn to reach the IC and then navigated straight to the PBS in WT brains (*Figure 4C*, left panels and *Figure 4—video 1*). In *Kif7⁻/⁻* embryos, most thalamic axons stopped their course after leaving the diencephalon and formed two short bundles: a large one oriented to the IC (*Figure 4C*, right panels, arrowheads and *Figure 4—video 2*) and another one oriented to the amygdala, more caudally and ventrally (*Figure 4C*, white arrows). This last projection was never observed in WT brains. Similar analyses performed at E16.5 showed that 2 days later, most TCA had reached the cortex in both WT and *Kif7⁻/⁻* embryos (*Figure 4D*, arrowheads) despite a small delay in *Kif7⁻/⁻* embryos and abnormal trajectories in the PSB region (*Figure 4D1 and D2*). The ectopic ventral caudal projection to the amygdala was still present at E16.5 in *Kif7⁻/⁻* embryos (*Figure 4D1*, arrow). At E18.5, the TCA projections in the *Kif7⁻/⁻* embryos appeared almost normal in the rostral cortex (*Figure 4E1*) but drastically reduced in the median and caudal cortex (*Figure 4E2, E3, and E4*, arrowheads). In addition, a thick bundle of thalamic axons followed an abnormal trajectory at the PSB (*Figure 4E2 and E3*, arrow). Neurofilament staining with Smi32 antibodies confirmed the strong reduction of projections in the cortex and IC of *Kif7⁻/⁻* embryos (*Figure 4F*).

## Kif7 invalidation alters the cortical distribution of cIN at E14.5

Because cIN are born in the basal forebrain and likely depend on contact/functional interactions with pioneer corticofugal projections for the first stages of their migration (*Métin and Godement, 1996*; *Denaxa et al., 2001*), we examined their distribution in the developing cortex of *Kif7⁻/⁻* animals. cIN are generated outside of the cortex, in the medial and caudal ganglionic eminences (MGE, CGE) and preoptic area (POA) of the ventral forebrain. They enter the lateral cortex at E12.5-E13.5, depending on mouse strains, and colonize the whole cortex by organizing two main tangential migratory streams, a superficial one in the MZ, and a deep and large one in the lower TBR2(+) IZ/SVZ (*Tanaka et al., 2003*; *Yokota et al., 2007*). We thus analyzed the cortical distribution of tdTomato(+) MGE-derived cIN in WT and *Kif7⁻/⁻* Nkx2.1-Cre;Rosa26-tdTomato transgenic embryos. An abnormal MGE cell distribution was observed in the developing cortex of *Kif7⁻/⁻* embryos from E14.5 (*Figure 5*, *Figure 5—figure supplement 1*). In mutant brains, both the latero-medial extent of the superficial and deep tangential migratory streams and the space separating them were significantly shortened compared to WT brains (*Figure 5A2, B2, and B3*). The closeness of the two streams in mutants was consistent with the reduced thickness of the corticofugal and TCA projections that navigate tangentially in the IZ (illustrated in *Figure 4*). The superficial migratory stream was thinner and denser in *Kif7⁻/⁻* embryos compared to WT (*Figure 5A1 and B1*, right panels). Remarkably, cIN no longer colonized the dorsal cortex of *Kif7⁻/⁻* embryos, and the deep tangential stream stopped in the region where the layer of TBR2(+) SVZ cells switched to the cortical surface (*Figure 5A1*, white arrow). The tangential progression of cIN in the developing cortex is controlled by CXCL12, a chemokine transiently expressed in TBR2(+) cells and in meninges (*Stumm et al., 2003*; *Li et al., 2008*). CXCL12 prevents the premature occurrence of the tangential to radial migration switch required for cIN to colonize the CP (*Tiveron et al., 2006*; *Stumm et al., 2003*; *Li et al., 2008*; *Atkins et al., 2023*). In situ hybridization (ISH) experiments revealed that *CxCl12* mRNA was not expressed in the dorsal cortex of E14.5 *Kif7⁻/⁻* embryos (*Figure 5C*, black arrow). To characterize the consequences of the abnormal expression pattern of CXCL12 at E14.5 on the migratory behavior of cIN, we sliced coronally E14.5 *Kif7⁻/⁻* and WT forebrains and imaged by time-lapse video-microscopy to track the trajectories of MGE-derived tdTomato-expressing cIN in the dorsal cortex of organotypic slices (*Figure 5D*). While most cIN presented tangential and oblique trajectories in WT slices, a majority of cIN exhibited radially oriented trajectories in *Kif7⁻/⁻* slices (*Figure 5D*). CXCL12 expression in SVZ strongly decreases at E16.5 (*Tiveron et al., 2006*; *Caronia-Brown and Grove, 2011*), and analyses performed at E16.5 in fixed *Kif7⁻/⁻* and WT embryos showed that cIN had pursued their migration and reached the dorsal cortex (*Figure 5—figure supplement 1A*) even though the tangential migration of cIN in *Kif7⁻/⁻* embryos remained delayed as compared to WT embryos. Analyses performed in a surviving P0 animal (*Figure 5—figure supplement 1B and C*) revealed an abnormal distribution of cIN in the CP characterized by a strong decrease of cIN density in the infragranular layers that were moreover much thinner than in control brains (see CTIP2 staining in *Figure 5—figure supplement 1C*).

The abnormal distribution of cIN in the cortex of *Kif7⁻/⁻* embryos reflected the structural and molecular abnormalities that we had identified in their developing cortex. We next examined if migratory defaults proper to *Kif7⁻/⁻* cIN could moreover contribute to their abnormal cortical distribution.

## Kif7 invalidation affects the migratory behavior of cIN in co-cultures and organotypic slices

We thus compared the migratory behavior of *Kif7⁻/⁻* and WT cIN using an in vitro model previously established in the lab (*Bellion et al., 2005*) to compare the dynamics of mutant and WT cIN migrating on a substrate of WT cortical cells (*Figure 6A*). WT and *Kif7⁻/⁻* E14.5 MGE explants from tdTomato-expressing embryos were cultured on WT dissociated cortical cells (*Figure 6A1*). Fluorescent migrating MGE cells were imaged using time-lapse video microscopy and tracked over time (*Figure 6A2*). Dynamic parameters (speed during moves, stops, *Figure 6A3–A6*) and trajectories (*Figure 6A7*) were analyzed on videos as illustrated on *Figure 6—videos 1–4*. *Kif7⁻/⁻* cIN migrated slightly more rapidly than WT cIN (*Figure 6A3*) mainly because the duration of their stops was significantly reduced (*Figure 6A4*). The saltatory behavior of cIN was preserved (same frequency of fast nuclear movements, *Figure 6A5*) despite a slight decrease in the speed of fast nuclear movements (*Figure 6A6*). WT cIN migrated along pretty straight trajectories (*Figure 6A2*, left panel and *Figure 6A7*, black curve), whereas *Kif7⁻/⁻* cIN showed reduced directionality persistence over time (*Figure 6A2*, right

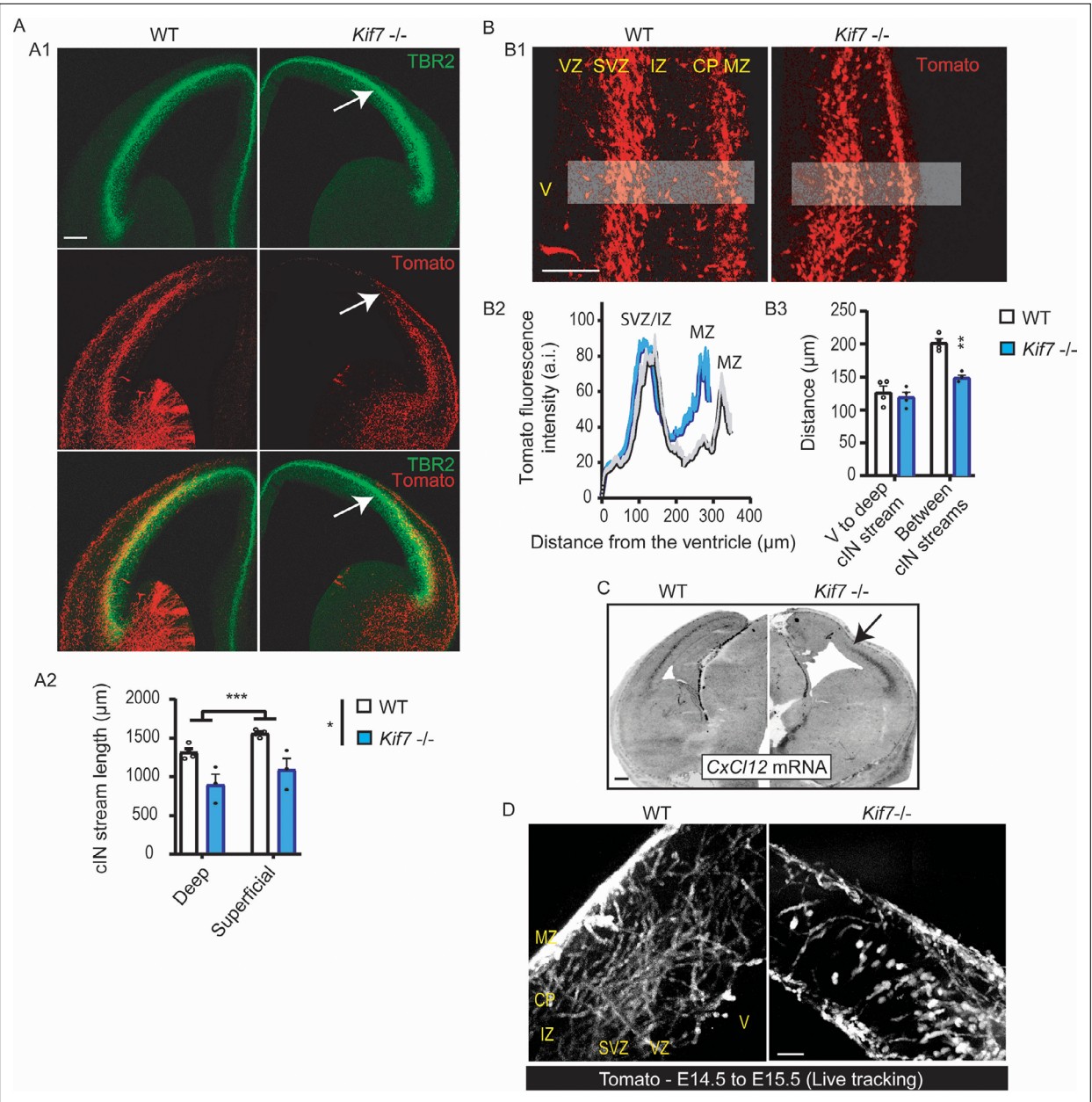

**Figure 5.** Abnormal cortical distribution of cIN and *Cxcl12* transcript expression in embryonic stage 14.5 (E14.5) in *Kif7⁻/⁻* brains. (**A**) The cortical distribution of cIN is visualized in wild-type (WT) and *Kif7⁻/⁻* mouse embryos crossed with the Nkx2.1-Cre/R26R-tdTomato strain in which medial ganglionic eminence (MGE)-derived cIN express the fluorescent marker tdTomato. Panels in A1 compare the distribution of tdTomato (+) cIN in WT and *Kif7⁻/⁻* cortical sections prepared at the same rostro-caudal level and in which SVZ is immunostained with TBR2 antibodies (green). Pictures show that the deep migratory stream of cIN terminates in *Kif7⁻/⁻* brains in the cortical region where the TBR2(+) layer reaches the cortical surface. Quantitative analysis of the mean length of the deep and superficial migratory streams measured from the entry in the pallium to the last detected cIN in the cortex is illustrated in graph A2. Statistical significance is assessed using two-way ANOVA (layers [WT, n=4; *Kif7⁻/⁻*, n=3; ***, p=0.001] and genotype [*, p=0.0157]). (**B**) Representative pictures (**B1**) of the deep and superficial tangential migratory streams of cIN in the lateral cortex of WT and *Kif7⁻/⁻* embryos. Pictures illustrate the decreased thickness of the superficial stream and the reduced distance between the deep-superficial streams in *Kif7⁻/⁻* embryos. Graph in B2 (WT, n=4; *Kif7⁻/⁻*, n=4) compares the distribution of the fluorescence intensity along a ventricle/MZ axis (see gray rectangles in B1) using the plot profile function of Fiji. Curves show no change in the distance between the ventricular wall and the deep cIN, but a significant reduction of the distance between the two streams in the *Kif7⁻/⁻* cortical sections as quantified on the graph B3 (WT, n=4; *Kif7⁻/⁻*, n=4; two-way ANOVA reveals a significant interaction between genotype and layer [p=0.0233] and multiple comparisons, a statistical difference between genotype only for the distance between the cIN streams [**, p=0.0051]). (**C**) Panels compare the distribution of *Cxcl12* mRNA in WT (left panel) and *Kif7⁻/⁻* (right panel) forebrain coronal sections at E14.5. The WT section shows *Cxcl12* transcript enrichment in a deep cortical layer already identified as the SVZ. In the *Kif7⁻/⁻* cortical section, the expression of *Cxcl12* transcripts is reduced to the lateral part of the SVZ. (**D**) Z-projections of 30 frames acquired during 12 hr in the dorsal cortex

*Figure 5 continued on next page*

Figure 5 continued

of living organoptypic forebrain slices representative of E14.5 WT and *Kif7⁻/⁻* embryos with tdTomato expressing cIN. *Kif7⁻/⁻* cIN were able to migrate dorsally but followed preferentially radially oriented trajectories. Scale bars: 200 μm.

The online version of this article includes the following figure supplement(s) for figure 5:

**Figure supplement 1.** cIN migration up to birth.

---

panel and *Figure 6A7*, blue curve). *Kif7* ablation thus affected the migratory behavior of cIN in a cell-autonomous manner, and the migratory defaults of *Kif7⁻/⁻* cIN evoked those of *Kif3a⁻/⁻* cIN whose primary cilium is nonfunctional (see Figure S6 in *Baudoin et al., 2012*).

We then analyzed the dynamic behavior of *Kif7⁻/⁻* cIN in the mutant cortex (*Figure 6B*). In WT Nkx2.1-Cre;Rosa26-tdTomato transgenic slices, fluorescent cIN migrated tangentially from the PSB to the dorsal cortex in two main streams, the MZ and the lower IZ/SVZ (*Figure 6B1*, *Figure 6—video 1*). All along these pathways, cIN sporadically operated a tangential to oblique/radial migration switch to colonize the CP. High cell density in the MZ prevented cell monitoring, and MZ cells were discarded from analyses. In the deep stream, a large proportion of imaged cells (59.3%) either maintained tangentially oriented trajectories (28.7%, red trajectories in *Figure 6B1 and B3*, left column of *Figure 6B2*) or reoriented to the CP along oblique or radial trajectories (30.6% green trajectories in *Figure 6B1 and B3*, left column in *Figure 6B2*). Most of these cIN reached the CP surface and remained there. A small proportion of cIN moved from the deep stream to the ventricular side of the slice (11.7%, blue trajectories in *Figure 6B1 and B3*, left column in *Figure 6B2*). A significant proportion (15.6%) migrated radially over the cortex thickness, inverting their polarity in the MZ or at the ventricular surface (pink trajectories in *Figure 6B1 and B3*, left column in *Figure 6B2*). Rare cIN in the VZ, deep stream, and CP moved very short distances or did not move (3.6%, dark colors in the left column of *Figure 6B2*). Finally, 8% cIN (orange) moved backward to the ventral brain, and 1.7% (gray) showed no specific directionality or layer specificity and were classified as chaotic (*Figure 6B1 and B3*, left column in *Figure 6B2*). The majority of cIN presented a characteristic saltatory behavior, alternating stops and fast moves (*Bellion et al., 2005*). The dynamics of cells recorded in the deep stream and/or moving to the CP is shown in *Figure 6B4–6* (left column on histograms).

The trajectories followed by cIN in the lateral/latero-dorsal cortex of *Kif7⁻/⁻* slices (where *CxCl12* mRNA was detected as shown in *Figure 5C*) dramatically differed from those described above (compare WT and *Kif7⁻/⁻* columns in *Figure 6B2*, schemes in *Figure 6B3* and *Figure 6—videos 1 and 2*). The number of cIN migrating tangentially in the deep migratory stream and/or reorienting to the CP dropped drastically (17.8% as compared to 59.3% in WT slices) to the benefit of immobile cells (35.7% as compared to 3.6% in WT slices) especially in the SVZ/lower IZ. The proportions of cIN moving to the ventricle or to the CP were similar (10.2% and 13.9%, respectively, as compared to 11.7% and 30.6% in WT slices), suggesting that the radial asymmetry of cortical slices was lost for *Kif7⁻/⁻* cIN. The same proportion of cIN migrated radially across the cortical layers as in WT slices (17.2% versus 15.6% in WT), but the proportion of cIN with chaotic trajectories strongly increased (7.8% versus 1.7% in WT slices). Another major change was a significant decrease in the migration speed (excluding immobile cIN) (*Figure 6B4*) with less frequent but longer stops in *Kif7⁻/⁻* than in WT slices (*Figure 6B5 and B*), confirming the dynamic parameters measured on co-cultures.

*Kif7* ablation has been shown to activate SHH transcriptional signaling by blocking the cleavage of GLI3 in GLI3-R repressor, or conversely to inhibit SHH transcriptional signaling by preventing the formation of GLI activators (GLI1/2) (*Cheung et al., 2009*). Our western blot analysis confirmed that the GLI3-R/GLI3-FL ratio dropped in the *Kif7⁻/⁻* cortex according to the expected inhibition of the GLI3 processing, a result not observed in the MGE. Since *Kif7⁻/⁻* cIN are born in the MGE and migrate in the cortex where SHH signaling is activated due to an abnormal GLI3 processing, we first examined whether the application of SHH on a WT forebrain slice could mimic the migratory defaults of *Kif7⁻/⁻* cIN in the mutant cortical slices. SHH application on control slices minimally affected the trajectories of cIN (*Figure 6B2* compare WT and SHH [WT] columns). An increased proportion of cIN migrated radially across the cortical thickness (29% compared to 15.6% in WT slices) at the expense of cIN migrating from the deep stream to the CP (20.8% instead of 30.6% in WT slices). Frequent radial movements directed to the VZ (*Figure 6B2 and B3* and *Figure 6—video 3*) did not evoke the migratory defaults of *Kif7⁻/⁻* cIN in the mutant slices. Moreover, the frequency of chaotic trajectories did not increase (gray, *Figure 6B2*) and the main dynamic alterations, a decreased speed (*Figure 6B4*)

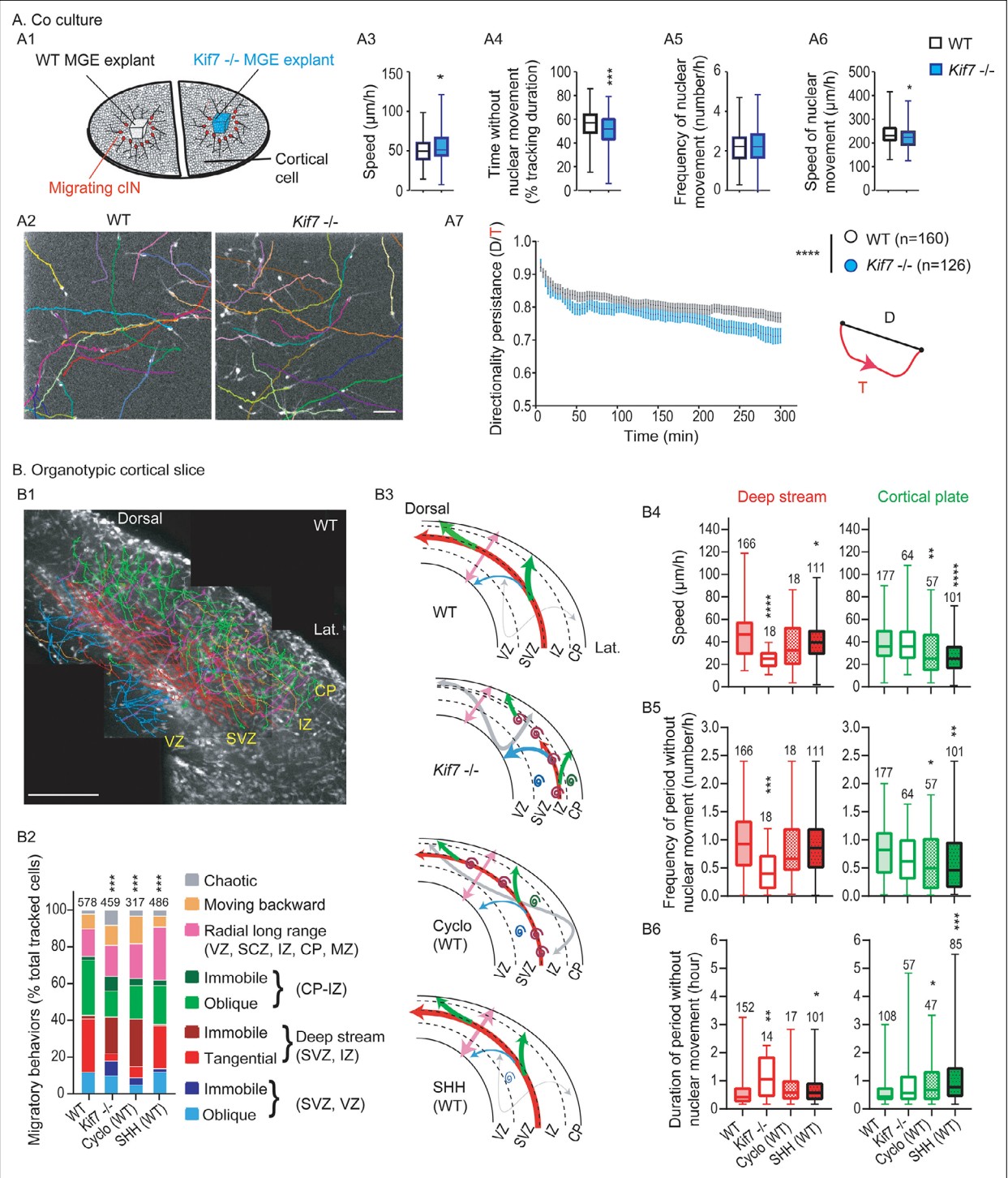

**Figure 6.** Dynamic behavior of migrating cIN in co-culture and organotypic cortical slices. (**A**) To prepare co-cultures, medial ganglionic eminence (MGE) explants were dissected out of telencephalic vesicles at embryonic stage 14.5 (E14.5) from wild-type (WT) or *Kif7⁻/⁻* Nkx2.1-Cre;Rosa26-tdTomato embryos and placed on a substrate of WT E14.5 dissociated cortical cells (**A1**). After 24 hr in culture, MGE cells migrated centrifugally away from MGE explants on the substrate of cortical cells and were recorded for 7 hr. WT (n=160) and *Kif7⁻/⁻* (n=126) MGE cells were tracked manually using the MTrackJ plugin allowing to analyze migratory parameters (**A2**). Box and whisker plots indicate the speed (**A3**), the time without nuclear movement (**A4**), the frequency (**A5**), and speed of nuclear movements (**A6**). Statistical significance is assessed by Mann-Whitney or unpaired tests. ***, p<0.001 (**A4**); *, p=0.0308 (**A3**), p=0.0123 (**A5**). Directionality persistence was calculated as the ratio of the cell displacement (distance between the first and last positions of the cell) on the cell trajectory (A7, scheme on the right). The graph represented the mean directionality ratios at each time point over the first 5 hr of recording for the recorded cells (**A7**). Statistical significance is assessed using two-way ANOVA and reveals a significant effect of

*Figure 6 continued on next page*

*Figure 6 continued*

time and genotype (****, p<0.0001). Scale bar: 50 µm (**A2**). (**B**) tdTomato(+) cIN migrating in living cortical slices from Nkx2.1-Cre;Rosa26-tdTomato embryos were tracked manually using the MTrackJ plugin and their trajectories color-coded as shown in legend (**B2**) to characterize their preferred direction (tangential, oblique, radial, immobile) and cortical layer localization (VZ-SVZ, IZ, CP). The picture in B1 illustrates the z-projection of trajectories reconstructed in a control slice, superimposed to the last picture of the video. Cortical interneurons migrating tangentially in the superficial migratory stream (MZ) could not be tracked because of high density. Graphs in B2 compare the percentage of each kind of trajectory recorded in the lateral cortex of control slices (control, see also *Figure 6—video 1*), *Kif7⁻ᐟ⁻* slices (*Kif7⁻ᐟ⁻*, see also *Figure 6—video 2*), control slices treated acutely with either murine SHH (SHH, see also *Figure 6—video 3*) or cyclopamine (Cyclo, see also *Figure 6—video 4*). Significance of the differences between the four distributions was assessed by a Chi-square test, $\chi^2$ (24, n=1224), p=0.0004998, ***. All experimental conditions differed from the control (Fisher's test, p=0.0004998, ***). Slices from three WT animals in control condition or treated with drugs and from three *Kif7⁻ᐟ⁻* animals were analyzed; the number of analyzed cells is indicated above bars. Schemes in B3 summarize the main results observed in each experimental condition. Trajectories are represented with the same color code as in B1, and line thickness is proportional to the percentage of cells exhibiting each type of trajectory. Immobile cells are figured by a coil. Box and whisker plots indicate the mean speed (**B4**), the frequency of period without nuclear movement (number/hr) (**B5**) and the mean duration of period without nuclear movement (hour) (**B6**) for cIN migrating tangentially in the deep stream (red box and whisker plots, left) or to the cortical plate (green box and whisker plots, right). Statistical significance assessed by Kruskal-Wallis tests in each cluster. ****, p<0.0001; ***, B5 left p=0.0005; **, B4 right p=0.0088, B5 right p=0.0037, B6 left p=0.0034; *, B4 left p=0.0233, B5 right p=0.0470, B6 left p=0.0134, B6 right p=0.0394. Number of analyzed cells is indicated on plots. VZ, ventricular zone; SVZ, subventricular zone; IZ, intermediate zone; CP, cortical plate; MZ, marginal zone. Scale bar: 300 µm.

The online version of this article includes the following video(s) for figure 6:

**Figure 6—video 1.** Migration of tdTomato-expressing Kif7⁻ᐟ⁻ medial ganglionic eminence (MGE) cells.
https://elifesciences.org/articles/100328/figures#fig6video1

**Figure 6—video 2.** Migration of tdTomato-expressing wild-type (WT) medial ganglionic eminence (MGE) cells treated with mouse Sonic Hedgehog (SHH).
https://elifesciences.org/articles/100328/figures#fig6video2

**Figure 6—video 3.** Migration of tdTomato-expressing wild-type (WT) medial ganglionic eminence (MGE) cells treated with mouse Sonic Hedgehog (SHH).
https://elifesciences.org/articles/100328/figures#fig6video3

**Figure 6—video 4.** Migration of tdTomato-expressing wild-type (WT) medial ganglionic eminence (MGE) cells treated with cyclopamine.
https://elifesciences.org/articles/100328/figures#fig6video4

associated with an increased duration of stops (*Figure 6B6*), were milder than those observed in *Kif7⁻ᐟ⁻* cIN. On the contrary, the application of the SHH pathway inhibitor cyclopamine on WT cortical slices altered cIN trajectories in a way that resembled alterations in *Kif7⁻ᐟ⁻* slices (compare WT and cyclo [WT] columns in *Figure 6B2* and scheme in *Figure 6B3*). For example, the proportion of immobile cIN and of cIN moving short distances reached 33.7% as compared to 3.6% in WT slices and 35.7% in *Kif7⁻ᐟ⁻* slices (*Figure 6B2*). Cyclopamine application mainly affected the ability of cIN to move tangentially in the deep stream and to switch to the CP (*Figure 6B2 and B3* and *Figure 6—video 4*). Nevertheless, the migration speed defects of *Kif7⁻ᐟ⁻* cIN and of cyclopamine-treated cIN differed (*Figure 6B4*).

In conclusion, *Kif7⁻ᐟ⁻* cIN exhibited intrinsic migratory defaults resembling those of cIN with a primary cilium ablation and showed cortical trajectories defaults resembling those of cyclopamine-treated WT cIN. *Kif7* ablation in cIN had the same effect as SHH signaling inhibition on cIN migratory behavior.

The strong influence of cyclopamine on the migration of cIN in WT cortical slices led us to determine the distribution of the endogenous activator SHH in the developing cortex.

## Distribution of Shh mRNA and SHH protein in the E14.5 cortex

*Shh* mRNA is strongly expressed in the mantle zone of the ganglionic eminences (MGE, CGE) and POA during the time cIN differentiate (*Xu et al., 2005*; *Kon et al., 2017*), but its expression level in cIN after they reach the cortex is controversial (*Baudoin et al., 2012*; *Komada, 2012*; *Sahara et al., 2007*). We thus reexamined *Shh* mRNA expression in the forebrain and SHH protein distribution in cortical layers. ISH experiments performed at E13 confirmed the strong expression of *Shh* mRNA in the ventral forebrain, especially in the SVZ and mantle zone of the MGE (*Figure 7A1*) in agreement with the SHH-dependent *Gli1* mRNA expression in a VZ subdomain of the MGE (*Nery et al., 2001*). In contrast, *Shh* transcripts were barely detectable in the cortex. Using RNAscope, which has higher sensitivity and spatial resolution than standard ISH, we re-examined the expression pattern of *Shh* mRNA at E14.5. In the ventral forebrain, RNAscope *Shh* transcript detection showed similar

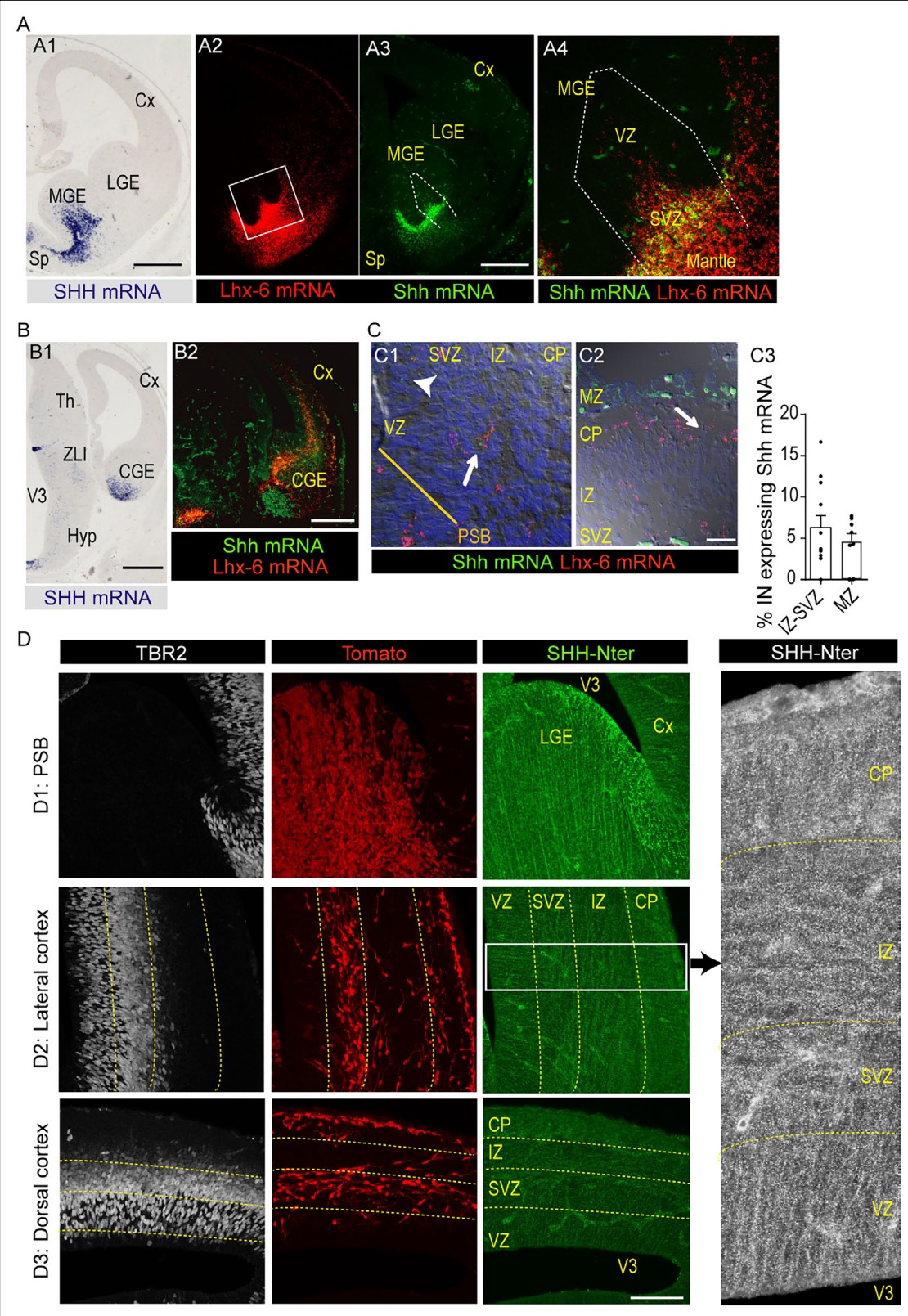

**Figure 7.** Expression of *Shh* transcripts. (**A–C**) and distribution of Sonic Hedgehog (SHH) protein (**D**) in the embryonic stage 13.5 (E13.5)-E14.5 forebrain. (**A, B**) Distribution in median (**A**) and caudal (**B**) coronal sections of Shh mRNA detected by in situ hybridization with an antisense *Shh* probe at E13.5 (**A1, B1**) and by RNAscope at E14.5 (**A2-3, B2**). *Shh* transcripts are strongly expressed in the medial ventral forebrain (SVZ and mantle zone of the MGE and septum, **A1, A2**), in the mantle zone of the CGE (**B1, B2**), in the zona limitans intrathalamica (ZLI in B1) and in the ventral midline of the third

*Figure 7 continued on next page*

*Figure 7 continued*

ventricle (V3 in B1). RNAscope further confirmed the strong expression of *Shh* mRNA (A3, green) in MGE and septum regions that strongly express *Lhx-6* mRNA (A2, red). Confocal observations in the SVZ and mantle zone of the MGE showed that *Shh* mRNA (green) is co-expressed with the *Lhx6* mRNA (red) in a significant number of cIN (yellow cells in A4). (**C**) Confocal analyses at higher magnification of the double detection by RNAscope of *Shh* and *Lhx-6* mRNA. Cells were identified on stacked images (△z=1 µm) using Nomarski optic. In the lateral cortex close to the PSB (C1, z-projection of 10 confocal planes) and in the dorsal cortex (C2, z-projection of 10 confocal planes), a very small proportion of cells expressing *Lhx-6* mRNA also express *Shh* mRNA (white arrows in C1,C2). Counting in the deep stream (SVZ-IZ) and in the MZ is shown in graph C3 (9–17 fields in three sections). A few progenitors in the cortical VZ express *Shh* mRNA at very low levels (arrowhead, **C1**). (**D**) SHH-Nter and TBR2 co-immunostaining of Nkx2.1-Cre/R26R-tdTomato brain coronal sections at E14.5. Representative confocal merged stacked images (△z=0.2 µm; 48 images) in the pallium-subpallium boundary (PSB, **D1**), lateral cortex (**D2**) and dorsal (**D3**) cortex revealing SHH-Nter immunostaining in blood vessels and the presence of numerous bright dots all over the cortical neuropile. On the ventricular side of the PSB and of the lateral-most part of the LGE, SHH-Nter(+) bright elements are aligned radially. In the lateral cortical neuropile, smaller bright dots align radially in the VZ, tangentially in the SVZ-IZ, and radially in the CP (see higher magnification on the right panel in which SHH-Nter immunostaining is shown in white and the contrast is increased). Cx, cortex; LGE, MGE, and CGE, lateral, medial, and caudal ganglionic eminence; CP, cortical plate; Hyp, hypothalamus; PSB, pallium-subpallium boundary; Sp, septum; Th, thalamus; V3, third ventricle; VZ, ventricular zone; SVZ, subventricular zone; IZ, intermediate zone; MZ, marginal zone. Scale bars: 500 µm (**A, B**), 20 µm (**C**), 100 µm (**D**).

The online version of this article includes the following figure supplement(s) for figure 7:

**Figure supplement 1.** SHH-Nter immunostaining specificity.

expression pattern as ISH (*Figure 7A1, A3, B1 and B2*). In addition, *Shh* mRNA and *Lhx-6* mRNA, a cIN marker, were expressed in overlapping domains in the mantle zone of the MGE (*Figure 7A2–4*). In the cortex, radially oriented cells in the VZ/SVZ and tangentially oriented cells in the deep migration stream expressed *Shh* mRNA at a very low level (one or two dots versus large cluster in the MGE) (*Figure 7C*). Among the large number of cells expressing *Lhx-6* transcripts (*Figure 7C1 and C2*), only 5% co-expressed *Shh* mRNA (*Figure 7C3*).

Antibodies that recognize the N-ter part of SHH (SHH-Nter, e.g. 'activated' SHH; *Lee et al., 1994*) strongly labeled in C57Bl/6 mouse brain the same regions known to strongly express SHH mRNA (*Sahara et al., 2007*; *Loulier et al., 2005*; *Kiecker and Lumsden, 2004*), such as the lateral wall of the third ventricle (*Figure 7—figure supplement 1A1*) and the zona limitans intrathalamica at E12.5 (*Figure 7—figure supplement 1A2*) and the choroid plexus and septum at E14.5 (*Figure 7—figure supplement 1B*). SHH-Nter immunostaining was detected in the cortex at E12.5 and E14.5 (*Figure 7—figure supplement 1C and D*). To further investigate the expression of SHH in the cortical wall at E14.5, we performed immunostaining experiments in Nkx2.1-Cre/R26R-tdTomato embryos (*Figure 7D*). SHH-Nter antibodies detected large bright dots that distributed radially away from the ventricle in the lateral LGE and at the ventricular angle (green dots in *Figure 7D1*, upper panel). In the cortex, SHH-Nter antibodies immunostained bright tiny dots that show remarkable distribution (*Figure 7D2 and D3*). In the lateral (*Figure 7D2*) and dorsal (*Figure 7D3*) cortex, the bright SHH-ter immunopositive dots were aligned radially in the VZ and in the CP according to the radial arrangement of cortical progenitors and CP cells. In contrast, dots were aligned tangentially in the SVZ/IZ where cIN distributed among tangentially oriented growing axons. The intensity and density of positive dots appeared lower in the dorsal than in the lateral cortex, suggesting a latero-dorsal gradient of distribution (compare *Figure 7D2 and D3*). Some CP cells presented a diffuse cytoplasmic staining, which was not observed in the bottom layers. Together, our ISH and immunostaining experiments showed that the cortical SHH protein probably has an extrinsic origin, as previously described at the early stage of cerebellum development (*Huang et al., 2010*).

## Discussion

Our aim in the present paper was to better understand the developmental origin of functional abnormalities (e.g. epilepsy, cognitive deficits, etc.) in cortical circuits of human patients with *KIF7* mutations by characterizing the developmental abnormalities of the cerebral cortex in a *Kif7*[-/-] mouse model. We show that *Kif7* ablation affects the development of the two populations of cortical neurons, CP neurons, and GABAergic interneurons, in specific and distinct ways, according to their dorsal and ventral origin in the telencephalon. The alterations of the CP development (lack of most subplate cells in the dorsal cortex and delayed corticofugal projection) are reminiscent of defaults previously described in GLI3 null mutants that lack most subplate neurons and pioneer cortical projections (*Theil, 2005*). The abnormal distribution of *Kif7*[-/-] cIN in the developing cortex partly reflects the structural

and functional alterations of cortical layers on which cIN migrate (e.g. abnormal CXCL12 expression in the SVZ). However, we identify additional migration defaults of $Kif7^{-/-}$ cIN that recall those of cIN lacking a primary cilium or of WT cIN treated with the SHH pathway inhibitor cyclopamine, suggesting that the SHH pathway activity is inhibited in $Kif7^{-/-}$ cIN. $Kif7$ ablation thus alters differently the SHH pathway activity in CP and cIN subpopulations, in agreement with their dorsal or ventral origin.

## $Kif7$ knockout mice as a model to study the effect of $KIF7$ mutations in human

KIF7 is a ciliary kinesin whose mutations in patients and ablation in mice alter the length rather than the structure of primary cilia. Primary cilium length was increased in patient fibroblasts (*Putoux et al., 2011*), in MEF isolated from $Kif7^{-/-}$ and in the neural tube of $Kif7^{-/-}$ embryos at E10.5 (*Lai et al., 2021*). Owing to the fact that KIF7 regulates the trafficking and processing of GLI factors, the cilium-dependent SHH signaling was altered in both patient fibroblasts (*Putoux et al., 2011*) and in $Kif7^{-/-}$ embryos (*Cheung et al., 2009*; *Lai et al., 2021*) with GLI3-R decrease and GLI1/GLI2 upregulation. These common features observed in cells from patients and in $Kif7^{-/-}$ mice models, as well as a previous study of the corpus callosum agenesis in $Kif7$ mutant mice, which helped to dissect the mechanism at work in patients (*Putoux et al., 2019*) allowed to propose that the $Kif7^{-/-}$ mouse is a good model to study the alterations that lead to clinical features in patients. Most $Kif7^{-/-}$ mice died at the end of gestation. However, the brain developed up to birth, allowing for the study of its development.

All patients carrying mutations in the $KIF7$ gene on both alleles have DD/ID and some have seizures. Investigations on cerebral malformation by MRI have shown that some of them have alterations in the cerebral cortex, including cortical atrophy (*Bakalinova, 1998*; *Ali et al., 2012*; *Walsh et al., 2013*) and pachygyria (*Tunovic et al., 2015*). Since detecting defects in the cerebral cortex is more demanding in MRI quality images than corpus callosum agenesis or MTS, we can speculate that the number of patients with cerebral cortex defects is underscored. Malformations of cortical development were recognized as causes of ID (*Marín, 2012*) and epilepsy (*Barkovich et al., 2015*), and the cellular defects underlying these pathologies could imply defects in cell proliferation, neuronal migration, and differentiation that lead to improper excitatory/inhibitory balance, connectivity into the cortex or of the cortex with other brain structures. The mechanism of action of KIF7 to transduce SHH ciliary-dependent signaling has been debated. It has been accepted that SHH activates KIF7 binding to microtubules and its accumulation to the tip of the cilium, making the kinesin motor domain the most essential domain for SHH transduction. A more recent study proposed that KIF7 is an immobile kinesin translocated to the cilium tip after SHH binding by a complex KIF3A/Kif3B/KAP (*Yue et al., 2022*), suggesting that domains in the KIF7 protein essential to its function are more complex. This could explain why any mutation, even point mutations in the coiled-coil domain or in the C-terminal part of the KIF7 protein, induces clinical features as severe as when mutations led to a highly truncated protein. Another explanation is that any mutation causes improper folding of the protein, probably inducing its degradation by the proteasome.

## KIF7 loss of function alters the cerebral cortex organization and the connectivity between the cortex and the thalamus

Our study demonstrates that in WT embryos, the levels of GLI3-FL and GLI3-R were much higher in the developing cortex than in the MGE, and the cleavage of GLI3-FL in GLI3-R was more important in the cortex. Our observations are coherent with previous studies that identified a preponderant role of GLI3-R signaling to pattern the dorsal telencephalon and to regulate the proliferation and differentiation of neurons in the cortex (*Wang et al., 2011*; *Wilson et al., 2012*). Moreover, the strong decrease of the cleavage of GLI3-FL in GLI3-R that we observed only in the cortex of $Kif7^{-/-}$ embryos suggests that the hyper SHH signaling resulting from this decreased cleavage affects the cortex, but not the MGE. The abnormal structure of the dorsal telencephalon in $Kif7^{-/-}$ embryos (enlarged lateral ventricles, cortical wall folding, and thinning) resembled the abnormal structure observed in ciliary mutants (*Magnani et al., 2015*; *Willaredt et al., 2008*) and in GLI3 mutants (*Magnani et al., 2010*; *Magnani et al., 2013*; *Theil, 2005*). As in $Kif7^{-/-}$ embryos, a mild PBS disorganization and an abnormal differentiation of the caudal telencephalic vesicle and telo-diencephalic boundary that is dependent on SHH and GLI3 for its formation (*Rash and Grove, 2011*) were also reported in these mutants (*Willaredt et al., 2008*; *Magnani et al., 2010*). The cellular disorganization observed in the developing cortex of

*Kif7*[-/-] embryos also recalls defects of *Gli3* mutants, especially the impaired preplate and SP neurons differentiation, the abnormal cortical layer formation, and abnormal axonal growth (*Magnani et al., 2010*; *Magnani et al., 2013*; *Theil, 2005*). Several of these traits have been observed in the cortex of the ciliary mutant Rfx3[-/-] that also exhibits a decreased GLI3-R/GLI3-FL ratio in the dorsal telencephalon (*Magnani et al., 2015*). Interestingly, the knockdown of KIF7 in E14.5 cortical cells altered several developmental stages of cortical neurons, leading to a decrease of apical radial progenitors, basal IP divisions, and to impaired neuronal polarity and migration (*Guo et al., 2015*). Sporadic heterotopia, as we observed in the cortex of *Kif7*[-/-] embryos, has been reported in one patient (*Tunovic et al., 2015*). Cortical heterotopia is not a landmark of ciliopathy but has been also described at E11 in the forebrain of the *Ift88* hypomorphic mutant *cobblestone* (cbs, *Willaredt et al., 2008*). And cellular alterations that should contribute to the formation of heterotopia, e.g., defective capacity to polarize and to migrate to the CP, were described in post-mitotic neurons electroporated with shRNA libraries targeting ciliary proteins such as KIF7, BBS1, BBS10, NPHP8, ALMS1 (*Guo et al., 2015*). Heterotopias were also detected in a Joubert syndrome patient carrying a mutation in the *CELSR2* gene coding for a planar cell polarity protein (*Vilboux et al., 2017*) essential for neural progenitor cell fate decision, neural migration, axon guidance, and neural maturation (*Boutin et al., 2012*; *Hakanen et al., 2022*).

Besides delayed growth of cortical and thalamic axons, *Kif7*[-/-] embryos exhibited connectivity defects between the thalamus and cortex, with abnormal trajectories of axon bundles in the basal forebrain, at the telo-diencephalic junction and along the PSB. To our knowledge, the ectopic projection of cortical axons to the pre-optic area had not been observed in GLI3 mutants that exhibited instead an ectopic thalamic projection to the amygdala (*Magnani et al., 2010*; *Magnani et al., 2013*). In the ciliary mutants *Rfx3*[-/-], ectopic cortical and thalamic axons were indeed observed in the basal telencephalon (*Magnani et al., 2015*). Cortical neurons electroporated in utero with shRNAs to deplete KIF7 or ciliary proteins belonging to the BBSome extended axons that showed abnormal fasciculation and trajectories (*Guo et al., 2015*), demonstrating a cell-autonomous role of ciliary genes to control the axonal pathfinding. Finally, a large population of thalamic axons that had reached the cortex of *Kif7*[-/-] embryos showed a massive elimination at E18.5 in the caudal telencephalon. This massive loss paralleled the loss of CP cells, as previously observed in Rfx3[-/-] and Inpp5e[-/-] mutants (*Magnani et al., 2015*). Although disorganized thalamocortical connectivity had never been reported in patients with *KIF7* mutation or in patients with ciliopathy, thalamocortical connectivity is essential for cognitive performances in young infants (*Alcauter et al., 2014*), and its alteration leads to neurodevelopmental delay and later cognitive deficits (*Jakab et al., 2020*). Moreover, a link between ciliary function, cortex development, and connectivity with the thalamus had been previously identified in mutant mice lacking the planar cell polarity proteins CELSR2 and FZD3. These mice display ciliary defects and an abnormal development of tracts between the thalamus and the cortex leading to a hypotrophy of the IZ where thalamocortical and corticofugal axons navigate (*Tissir et al., 2005*; *Wang et al., 2002*).

## Extra-cortical origin of SHH

Besides its major function in forebrain patterning (*Chiang et al., 1996*; *Rallu et al., 2002*), SHH regulates the embryonic and perinatal cerebral cortex development (*Komada et al., 2008*; *Dahmane et al., 2001*). The origin of the morphogen in this brain structure remains obscure. In agreement with a recent set of data from *Moreau et al., 2021*, we showed that cIN born in the MGE or POA no longer express SHH after they reach the developing cortex. In addition, the level of local *Shh* mRNA synthesis was extremely low in the cortex at E14.5. In contrast, SHH protein was detected in immunopositive dots distributed throughout the cortex. Dots aligned radially in the VZ and CP and tangentially in the IZ. This cortical distribution is compatible with an extrinsic origin of SHH and a transventricular delivery as described in the cerebellar VZ (*Huang et al., 2010*). SHH is indeed present in the cerebrospinal fluid and likely secreted by the choroid plexus, since *Shh* mRNA is expressed in the choroid plexus of mice from E12 to E14.5 (*Zakaria et al., 2019*). Accordingly, we showed here that the choroid plexus of the lateral ventricles was strongly immunopositive for SHH at E14.5. Interestingly, the CDO/BOC co-receptor - that favors SHH transport - was also detected along radially oriented structures by immunostaining in the mouse embryonic cortex (not illustrated). SHH could thus be transported away from the VZ and/or meninges in vesicular structures able to diffuse across the cortical thickness. Accordingly, it has been shown that SHH transport needs ESCRT-III member protein CHMP1A (*Ibisler et al., 2015*) and SHH-related proteins are found in exosomes secreted from *Caenorhabditis elegans*

epidermal cells (*He et al., 2014*), *Drosophila* wing imaginal discs (*Wang et al., 2002*), chick notochord cells, and human cell lines (*Vyas et al., 2014*). How SHH is transported throughout the cortex should be further explored. It has already been shown that SHH is present in multivesicular bodies, in neurites and filopodia in the postnatal hippocampus and cerebellum, and in vesicles located inside and outside of hippocampal neurons in culture (*Eitan et al., 2016*).

## Kif7 deletion leads to defects in cortical interneuron migration

The defects induced by gene mutations responsible for primary cilium dysfunction, including *Kif7* deletion, have been examined in PNs born in the dorsal telencephalon, which organize the cortical layers (*Hasenpusch-Theil et al., 2020*; *Higginbotham et al., 2013*). However, the cIN are born ventrally in the MGE where the expression levels of GLI3-R and GLI3-FL are extremely low in both WT and *Kif7*[-/-] embryos. cIN leave the MGE and then migrate to the cortex during the embryonic development. Their primary cilium is functional and can transduce local SHH signals to influence their directionality, especially the tangential to radial reorientation of their trajectories toward the CP (*Baudoin et al., 2012*). In the present study, we showed that the abnormal distribution of cIN in the mutant cortex of *Kif7*[-/-] embryos resulted from cell-autonomous defects and from extrinsic cortical defects, especially the transient lack of CXCL12 expression in the dorsal cortex at E14.5. Interestingly, *Kif7* ablation affected the migratory behavior of cIN cultured on a WT cortical substrate in a similar way as did the ablation of *Kif3a* (*Baudoin et al., 2012*), another ciliary gene required for the transduction of SHH signals (*Han et al., 2008*). Accordingly, cyclopamine application on WT cortical slices altered the migratory behavior of cIN in a way that mimicked the migratory behavior of *Kif7*[-/-] cIN. We thus concluded from these observations that *Kif7* ablation inhibited the Shh pathway activity in cIN. By consequence, *Kif7*[-/-] cIN, which normally differentiates in a ventral telencephalic domain expressing high SHH level, presents dynamic and trajectories alterations that mimic those of cIN lacking SHH signaling. Therefore, *Kif7* ablation in MGE-derived cIN does not activate SHH signaling by decreasing the GLI3-R/GLI3-FL ratio, as observed in the dorsal telencephalon. On the contrary, Kif7 ablation in MGE-derived cIN likely inhibits SHH signaling by a mechanism independent of GLI3 processing that was unchanged in the MGE of *Kif7*[-/-] embryos (*Figure 2*). In addition to this cell-autonomous defect that impaired cIN migration, we show that the lack of CXCL12 expression in the dorsal cortex transiently and locally modified the directionality of *Kif7*[-/-] cIN, thereby altering the dorsal cortex colonization by cIN in *Kif7* mutants (*Tiveron et al., 2006*; *Lysko et al., 2011*).

Interestingly, acute SHH application on WT cortical slices slightly increased the percentage of radially migrating cIN. In contrast, cyclopamine that inhibited the SHH pathway blocked the migration of a large population of cIN in the deep cortical layers, demonstrating that endogenous cortical SHH is likely to activate the SHH pathway in migrating cIN to control their trajectories.

In conclusion, using the *Kif7*[-/-] murine model, we show that the *Kif7* deletion is responsible for a large range of developmental defects in the cerebral cortex, which include: (i) alteration of the inhibitory interneurons' migration by a combination of intrinsic and extrinsic factors, (ii) abnormal formation of cortical layers by the CP neurons, and (iii) major pathfinding defects in axonal projections between the cortex and the thalamus. Patients carrying mutations in the *KIF7* gene are classified as ciliopathic patients and display DD, ID, and epilepsy. These clinical features could thus be caused by an abnormal distribution of excitatory and inhibitory cortical neurons leading to impaired excitatory/inhibitory balance, and by the development of abnormal connections in the cerebral cortex and with other brain structures. By demonstrating focal heterotopia and abnormal thalamocortical connectivity in the murine model, we hope to open a new field of clinical investigations using MRI and tractography to identify such defects in patients carrying mutations in the *KIF7* gene.

## Materials and methods

### Mice

E14.5 embryos of the Swiss (for ISH and RNAscope) and C57Bl/6J (for SHH immunostaining) strains were obtained from pregnant females purchased at JanvierLabs (France). Male and female WT and *Kif7*[-/-] animals (E12.5, E14.5, E16.5, E18.5, and P0) expressing tdTomato or not in the MGE-derived cIN (Nkx2.1-Cre;Rosa26-tdTomato) were generated in our animal facility by mating male *Kif7*[+/-]; Nkx2.1-Cre;Rosa26-tdTomato and female *Kif7*[+/-]mice to produce embryos and P0. Thirty-four

pregnant females were used. *Kif7⁺/⁻* mice are a generous gift of Pr Chi-Chung Hui (the Provider, Sick-Kids Hospital, Toronto, USA) and Prof Bénédicte Durand (the Transferor, CNRS, Lyon, France). Mice were genotyped as described previously (*Cheung et al., 2009*). The day of the vaginal plug was noted E0.5. Experiments have been validated and approved by the Ethical Committee Charles Darwin (C2EA-05, authorized project 02241.02), and mice were housed and mated according to European guidelines. Both male and female animals were used in this study. Sex is not considered a biological variable in embryos.

## Brain lysate preparation and immunoblotting

Cortex and MGE of WT and *Kif7⁻/⁻* embryos from 5 litters were dissected at E14.5 and frozen at –80°C until use. Samples were sonicated in 80 µl Laemmli buffer and heated for 10 min at 37°C after adding 2.5% β-mercapto ethanol. Proteins were loaded on SDS-PAGE 7% gels (Bio-Rad) and transferred to 0.45 µm nitrocellulose membranes (Bio-Rad). Membranes were cut to detect signals from proteins above and below 55 kDa and were blocked with 50 g/l non-fat dry milk in PBS/0.1% Tween 20 for 1 hr at room temperature (RT), incubated with primary antibodies in the same solution overnight at 4°C, 1 hr at RT with appropriate IRDye-conjugated secondary antibodies, and imaged and quantified using ChemiDocMP Imaging System (Bio-Rad). The following commercial antibodies were used: anti GLI3 (R&D System-Biotechne, goat, #AF369, 1:200), anti-actin (Millipore, mice, #MAB1501, 1:10.000), Donkey-anti-goat-IR800 (Advansta, R-05781, 1:20.000), Goat-anti-mouse-StarBight Blue 700 (Bio-Rad, 12004159, RRID:AB_2884948, 1:2.500). Band intensities were quantified using gel analysis of Fiji software.

## Immunohistochemistry and imaging

Heads of embryos (from 5 to 10 litters depending on stage) and P0 animals (from 2 litters) were fixed by immersion in 0.1% picric acid/4% PFA in phosphate buffer (PB) for 4 hr and then in 4% PFA in PB overnight. After PBS washes, brains were dissected, included in 3% type VII agarose (Sigma, A0701), and sectioned at 70 µm with a vibratome. Immunostaining was performed on free-floating sections as explained previously (*Baudoin et al., 2012*). Primary antibodies were goat anti SHH-Nter (1:100, R&D Systems AF464, RRID:AB_355373), goat anti Netrin G1a (NG1a) (1:100, R&D Systems AF1166, RRID:AB_2154822), rabbit anti TBR1 (1:1000, Abcam ab31940, RRID:AB_2200219), rabbit anti TBR2 (1:1000, Abcam ab23345, RRID:AB_778267), rabbit anti PAX6 (1:100, clone poly19013, Covance PRB-278P, RRID:AB_291612), rabbit anti GSH2 (1:2000, Millipore ABN162, RRID:AB_11214376), chicken anti MAP2 (1:500, Novus, NB30213), mouse anti SMI-32 (1:500, Covance SMI-32R, RRID:AB_509997), and rat CTIP2 (1:1000, Abcam ab18465, RRID:AB_2064130). Primary antibodies were incubated in PGT (PBS/gelatin 2 g/l/Triton X-100 0.25%) with 0.1% glycine and 0.1% lysine for anti Netrin G1a antibodies and 3% BSA for anti SHH-Nter antibodies. Primary antibodies were revealed by immunofluorescence with the appropriate Alexa dye (Molecular Probes) or Cy3 (Jackson laboratories) conjugated secondary antibodies diluted in PGT (1:400). Bisbenzimide (1/5000 in PBT, Sigma) was used for nuclear counterstaining. Sections were mounted in mowiol/DABCO (25 mg/ml) and imaged on a macroscope (MVX10 olympus) or an epifluorescence microscope (LEICA DM6000) using X40 and X63 immersion objectives, or a confocal microscope (Leica TCS SP5) using a x40 objective. Double immunostaining was performed when possible, and images were representative of what was observed in more than four animals per genotype, except for heterotopia. Macroscope images were treated to remove background around sections using the wand tool in Adobe Photoshop 7.0 software. On epifluorescence and confocal microscope images, acquisition parameters and levels of intensity in Adobe Photoshop 7.0 software were similar for WT and *Kif7⁻/⁻*. Merged images were acquired on the same section except for TBR1 with TBR2 acquired on adjacent sections. The fluorescence intensity along the depth of the cortex was assessed using the plot profile function of ImageJ under a 500 pixels wide line starting in the ventricle to the surface of the brain.

## In situ hybridization

Specific antisense RNA probe for *Shh*, *Gli1*, and *CxCl12* genes (gift of Marie-Catherine Tiveron) was used for ISH analyses. DIG-probes were synthesized with a labeling kit according to the manufacturer's instructions (Roche, France). ISH was performed according to *Tiveron et al., 1996*, with few modifications. Fixed embryos (from 5 litters) were cryoprotected overnight in PBS with 10% sucrose,

embedded in OCT (Tissue-Tek, Miles, Elkhart, IN, USA) and frozen on dry ice. 10 μm cryostat sections were thaw-mounted on Superfrost slides (Menzel-Gläser), left to dry at RT, and stored at –80°C. Thawed sections were treated two X 10 min in RIPA buffer (150 mM NaCl, 1% NP-40, 0.5% Na deoxycholate, 0.1% SDS, 1 mM EDTA, 50 mM Tris, pH 8.0), postfixed in 4% PFA for 10 min at RT, and washed three X 5 min with PBS. The slides were then transferred in 100 mM triethanolamine, pH 8.0 for 2 min, and then acetylated for 10 min at RT by adding dropwise acetic anhydride (0.25% final concentration) while being rocked, and washed again three X 5 min in PBS. The slides were prehybridized briefly with 500 μl of hybridization solution (50% formamide, 5x SSC, 5x Denhardt's, 500 mg/ml herring sperm DNA, 250 mg/ml yeast RNA) and hybridized overnight at 70°C with the same solution in the presence of the heat-denatured DIG-labeled RNA probe. The following day, slides were washed in posthybridization solution (50% formamide, 2x SSC, 0.1% Tween-20) at 70°C first until coverslips slid off, then twice for 60 min at 70°C and finally at RT for 5 min. Slides were washed with buffer 1 (100 mM maleic acid, pH 7.5, 150 mM NaCl, 0.05% Tween 20), blocked for 30 min in buffer 2 (10% heat-inactivated horse serum in buffer 1), incubated overnight at 4°C with alkaline phosphatase-coupled anti-DIG antibody (Roche Diagnostics, Mannheim, Germany) diluted 1:1000 in buffer 2, rinsed twice for 5 min with buffer 1, and equilibrated for 30 min in buffer 3 (100 mM Tris, pH 9.5, 100 mM NaCl, 50 mM MgCl$_2$). The signal was visualized by a color reaction using 250 μl of buffer 4 per slice (6.6 μl/ml NBT [4-nitroblue tetrazolium chloride, Roche Diagnostics], 3.3 μl/ml BCIP [5-bromo-4-chloro-3-ind oyl-phosphate, Roche Diagnostics] in buffer 3). The color reaction was allowed to develop in the dark at RT during a few hours and was stopped with PBS. Sections were mounted on glass slides, dried, dehydrated in graded ethanol solutions, cleared in xylene, and coverslipped with Eukitt.

## RNAscope

Heads of E14.5 embryos (from 4 litters) were cut in cold Leibovitz medium (Invitrogen) and immersion fixed in cold 4% (wt/vol) PFA in 0.12 m PB, pH 7.4, overnight. Brains were then cryoprotected in PFA 4%/sucrose 10%, embedded in gelatin 7.5%/sucrose 10% at 4°C and frozen. Biological samples were kept at –80°C until coronally sectioned at 20 μm with cryostat. The RNAscope experiment was performed according the manufacturer's instructions (RNAScope Multiplex Fluorescent V2 Assay, ACDbio) after a pre-treatment to remove gelatin. Two different probes/channels were used (C1 for Shh, C2 for Lhx-6). Probes were diluted according to the manufacturer's instructions (1 vol of C2 for 50 vol of C1) and dilution of the fluorophores (Opal 520 and 570) was 1:1500. Sections were mounted in mowiol/DABCO (25 mg/ml) and imaged on a confocal microscope (Leica TCS SP5) using an immersion X10 or X63 objectives on 10 μm stacks of images (1 image/μm).

## DiI experiments

Tiny crystals of DiI (1,1'-dioctadecyl-3,3,3'3'-tetramethylindocarbocyanine perchlorate) were inserted in the dorsal or lateral cortex of WT (n=3) and Kif7$^{-/-}$ (n=4) embryonic brains fixed at E14.5 by immersion in 4% PFA. Injected brains were stored in 4% PFA at RT in the dark for several weeks to allow DiI diffusion. Before sectioning, brains were included in 3% agar. Coronal sections 60 μm thick were prepared with a vibratome and collected individually in 4% PFA. Sections were imaged with a macroscope (MVX10 Olympus).

## Whole-brain clearing and imaging

E14 and E16 embryos (WT, n=5; Kif7$^{-/-}$, n=6) were collected in cold L15 medium, perfused transcardially with 4% PFA using a binocular microscope. Brains were dissected and postfixed 3 days in 4% PFA at 4°C, and stored in PB at 4°C until clearing. All buffer solutions were supplemented with 0.01% sodium azide (Sigma-Aldrich). Whole-brain immunostaining and clearing was performed at RT under gentle shaking according to a modified version of the original protocol (*Renier et al., 2016*). Briefly, perfused brains were dehydrated in graded methanol solutions (Fischer Scientific) (20%, 50%, 80% in ddH$_2$O, 2×100%, 1h30 each), bleached overnight in 3% hydrogen peroxide (Sigma-Aldrich) in 100% methanol and rehydrated progressively. After PBS washes, brains were blocked and permeabilized 2 days in PBGT (0.2% gelatin [Merck], 0.5% Triton X-100 [Sigma-Aldrich] in PBS). Brains were incubated 3 days at 37°C in primary goat anti Netrin G1a (1:100, R&D Systems AF1166) and rabbit anti TBR1 (1:1000, Abcam ab31940) antibodies. After 1 day of PBGT washes, brains were incubated 1 day at 37°C in secondary 647 donkey anti-goat (1:1000, Molecular Probes,) and Cy3 donkey

anti-rabbit (1:1000, Jackson Laboratories) antibodies. Immunolabeled brains were washed 1 day in PBS, embedded into 1.5% low-melting agarose (type VII, Sigma) in 1% ultra-pure Tris-acetate-EDTA solution, placed in 5 ml tubes (Eppendorf, 0030119452), and dehydrated 1 hr in each methanol baths (50%, 80%, and 2×100%). Samples were then incubated for 3 hr in 33% methanol/66% dichloromethane (DCM, Sigma-Aldrich), washed in 100% DCM (2×30 min), and incubated overnight (without shaking) in dibenzyl ether (DBE). Brains were stored in DBE at RT. Cleared samples were imaged on a light-sheet microscope (LaVision Biotec, ×6.3 zoom magnification) equipped with an sCMOS camera (Andor Neo) and Imspector Microscope controller software. Imaris (Bitplane) was used for 3D reconstructions, snapshots, and videos.

## Brain slices

Brains of E14.5 WT (n=4 for control slices and n=4 for each pharmacological treatment) and *Kif7⁻/⁻* (n=3) transgenic Nkx2.1-cre, tdTomato embryos were dissected in cold L15 medium, embedded in 3% type VII agarose, and sectioned coronally with a manual slicer as explained in *Baudoin et al., 2012*. Forebrain slices 250 µm thick were transferred in Millicell chambers and cultured for 4 hr in F12/DMEM supplemented with CFS 10% in a $CO_2$ incubator (Merck Millipore) prior to pharmacological treatment and recording. Slices were incubated in drug for 2 hr before transfer in culture boxes equipped with a bottom glass coverslip for time-lapse imaging.

## Pharmacological treatment

Either recombinant Mouse SHH (C25II), N-terminus (464-SH-025, R&D Systems) at 0.5 µg/ml final or cyclopamine (C4116, Sigma) at 2.5 µM final were added to the culture medium of slices and renewed after 12 hr. Recombinant SHH was reconstituted at 100 µg/ml in sterile PBS with BSA 0.1%. The stock solution of cyclopamine was 10 mM in DMSO, and the culture medium of control experiments contained 1/4000 DMSO (vehicle). Control experiments in culture medium with and without vehicle did not differ and were analyzed together.

## Co-cultures

Co-cultures were performed on polylysine/laminin-coated glass coverslips fixed to the bottom of perforated Petri dishes in order to image migrating MGE cells. Brains were collected in cold PBS at E14.5. Cortices and MGE explants were then dissected in cold Leibovitz medium (Invitrogen). WT cortices were mechanically dissociated. Dissociated cortical cells were cultured on the coated glass coverslips and left in the incubator (37°C, 5% $CO_2$) for 1 hr. A wound divided the substrate in two halves on which WT and *Kif7⁻/⁻* MGE explants from littermate embryos were placed (*Figure 6A*). Co-cultures were cultured for 24 hr and four co-cultures were imaged.

## Videomicroscopy and image processing

Slices and co-cultures were imaged on an inverted microscope (Leica DMI4000) equipped with a spinning disk (Roper Scientific, USA) and a temperature-controlled chamber. Multi-position acquisition was performed with a Coolsnap HQ camera (Roper Scientific, USA) to allow the recording of the whole cortex. Images were acquired with a X20 objective (LX20, Fluotar, Leica, Germany) and a 561 nm laser (MAG Biosystems, Arizona). Z-stacks of 30 µm were acquired 50 µm away from the slice surface, with a step size of 2 µm and a time interval of 2 or 5 min for at least 21 hr. Acquisitions were controlled using the Metamorph software (Roper Scientific, USA). Cell trajectories were reconstructed on videos by tracking manually the cell rear with MTrakJ (ImageJ plugin, NIH, USA) and were clustered according to their position in cortical layers and to their orientation (see details in *Figure 6*). The migration speed of cells, the frequency, and duration of pauses were extracted from tracking data using Excel macros. The directionality of cells was analyzed along time using a macro from *Gorelik and Gautreau, 2014*.

## Study design

WT group was compared to *Kif7⁻/⁻* group and to pharmacological treatment groups for brain slice experiments. The experimental unit was a single animal for immunohistochemistry analysis and individual cell for videomicroscopy analysis.

## Statistical analysis

All data were obtained from at least three independent experiments and are presented as mean ± SEM (standard error of mean). Statistical analyses were performed with the GraphPad Prism software or R. Statistical significance of the data was evaluated using the unpaired two-tailed t-test, the Mann-Whitney test, the Chi-square test, or the two-way ANOVA test followed by a post hoc test. Data distribution was tested for normality using the D'Agostino and Pearson omnibus normality test. Values of $p < 0.05$ were considered significant. In figures, levels of significance were expressed by * for $p < 0.05$, ** for $p < 0.01$, *** for $p < 0.001$, and **** for $p < 0.0001$.

## Acknowledgements

Marie-Christine Tiveron is acknowledged for the gift of the CXCL12 expression vector. We thank all members of the Métin's Team for constructive discussions. We gratefully acknowledge the Imaging plateformplatform of the Fer à Moulin Institute for the use of their microscopes, and the animal facility of the Fer à Moulin Institute for animal care and breeding. This work was supported by Institut National de la Santé et de la Recherche Médicale (INSERM), Centre National de la Recherche Scientifique (CNRS), Sorbonne University. Agence Nationale de la Recherche (ANR, grant MIGRACIL to CM and ANR-23-IAHU-0010, France 2030 Program to JM), Fondation pour la Recherche sur le Cerveau (grant R11080DD to CM), Fondation J Lejeune (grant R14108DD to CM), DIM C-BRAINS.

## Additional information

### Funding

| Funder | Grant reference number | Author |
|---|---|---|
| Institut National de la Santé et de la Recherche Médicale | Recurring budget | Christine Metin |
| Centre National de la Recherche Scientifique | Salary | Justine Masson |
| Sorbonne Université | Recurring budget | Christine Metin |
| Agence Nationale de la Recherche | MIGRACIL | Christine Metin |
| Fondation pour la Recherche sur le Cerveau | R11080DD | Christine Metin |
| Fondation Jérôme Lejeune | R14108DD | Christine Metin |
| Agence Nationale de la Recherche | ANR-23-IAHU-0010 | Justine Masson |

The funders had no role in study design, data collection and interpretation, or the decision to submit the work for publication.

### Author contributions

María Pedraza, Sophie Scotto-Lomassese, Formal analysis, Investigation, Writing – original draft; Valentina Grampa, Investigation, Writing – original draft; Julien Puech, Sophie Lebon, Investigation; Aude Muzerelle, Azka Mohammad, Formal analysis, Investigation; Nicolas Renier, Formal analysis, Methodology; Christine Metin, Conceptualization, Formal analysis, Supervision, Funding acquisition, Writing – original draft, Project administration, Writing – review and editing; Justine Masson, Conceptualization, Formal analysis, Supervision, Investigation, Methodology, Writing – original draft, Project administration, Writing – review and editing

### Author ORCIDs

Nicolas Renier ⓘ https://orcid.org/0000-0003-2642-4402
Christine Metin ⓘ https://orcid.org/0000-0002-5936-0110
Justine Masson ⓘ https://orcid.org/0000-0002-4936-3334

## Ethics

Experiments have been validated and approved by the Ethical committee Charles Darwin (C2EA-05, authorized project 02241.02) and mice were housed and mated according to European guidelines.

Reviewer #1 (Public review): https://doi.org/10.7554/eLife.100328.4.sa1
Reviewer #2 (Public review): https://doi.org/10.7554/eLife.100328.4.sa2
Author response https://doi.org/10.7554/eLife.100328.4.sa3

# Additional files

## Supplementary files
MDAR checklist

Source data 1. Quantitative data used in the graphs and for statistical analyses (excel folder).

## Data availability

All data generated and analyzed during this study are included in the main manuscript and its supplementary materials. Uncropped blots have been supplied for Figure 2 (Figure 2—source data 1 and Figure 2—source data 2).

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
