## [Editor Report · eLife Assessment]

This **important** study provides **convincing** evidence that the Kinesin protein family member KIF7 regulates the development of the cerebral cortex and its connectivity and the specificity of Sonic Hedgehog signaling by controlling the details of Gli repressor vs activator functions. This study provides new insights into general aspects of cortical development.

---

## [Referee Report · Reviewer #1 (Public review)]

Summary:

This is an interesting follow-up to a paper published in Human Molecular Genetics reporting novel roles in corticogenesis of the Kif7 motor protein that can regulate the activator as well as the repressor functions of the Gli transcription factors in Shh signalling. This new work investigates how a null mutation in the Kif7 gene affects the formation of corticofugal and thalamocortical axon tracts and the migration of cortical interneurons. It demonstrates that Kif7 null mutant embryos present with ventriculomegaly and heterotopias as observed in patients carrying KIF7 mutations. The Kif7 mutation also disrupts the connectivity between cortex and thalamus and leads to an abnormal projection of thalamocortical axons. Moreover, cortical interneurons show migratory defects that are mirrored in cortical slices treated with the Shh inhibitor cyclopamine suggesting that the Kif7 mutation results in a down-regulation of Shh signalling. Interestingly, these defects are much less severe at later stages of corticogenesis.

Strengths/weaknesses:

The findings of this manuscript are clearly presented and are based on detailed analyses. Using a compelling set of experiments, especially the live imaging to monitor interneuron migration, the authors convincingly investigate Kif7's roles and their results support their major claims. The migratory defects in interneurons and the potential role of Shh signalling present novel findings and provide some mechanistic insights but rescue experiments would further support Kif7's role in interneuron migration. Similarly, the mechanism underlying the misprojection which has previously been reported in other cilia mutants remains unexplored. Taken together, this manuscript makes novel contributions to our understanding of the role of primary cilia in forebrain development and to the aetiology of the neural symptons in ciliopathy patients.

Comments on revisions:

The authors addressed most of the points I raised in my original review.

---

## [Referee Report · Reviewer #2 (Public review)]

Summary:

This study investigates the role of KIF7, a ciliary kinesin involved in the Sonic Hedgehog (SHH) signaling pathway, in cortical development using Kif7 knockout mice. The researchers examined embryonic cortex development (mainly at E14.5), focusing on structural changes and neuronal migration abnormalities.

Strengths:

(1) The phenotype observed is interesting, and the findings provide neurodevelopmental insight into some of the symptoms and malformations seen in patients with KIF7 mutations.

(2) The authors assess several features of cortical development, including structural changes in layers of the developing cortex, connectivity of the cortex with thalamus, as well as migration of cINs from CGE and MGE to cortex.

Comments on revisions:

The authors have made significant and thoughtful responses as well as experimental additions to the authors comments. Their efforts are appreciated and the manuscript is much improved.

---

## [Author Response]

The following is the authors’ response to the previous reviews.

**Reviewer #1 (Recommendations for the authors):**
(1) I am not convinced by the figures the authors present on Shh protein expression. The "bright tiny dots" of Shh protein in the cortex are not visible on the images in Figure 7. I wonder whether the authors could present higher magnification and/or black and white images with increased contrast.

We have modified Figure 7: we now present a higher magnification and a black and white image with increased contrast to better visualize SHH (+) bright tiny dots in the lateral cortex.

(2)The manuscript also contains several typos.

We apologize for these mistakes which have all been corrected.